

# A single-beam photothermal interferometer for in-situ measurements of aerosol light absorption

Bradley Visser[1], Jannis Röhrbein[1], Peter Steigmeier[1], Luka Drinovec[2,3], Griša Močnik[2,3,4], Ernest Weingartner[1]

[1] University of Applied Sciences Northwestern Switzerland, Windisch, Switzerland
[2] Jozef Stefan Institute, Ljubljana, Slovenia
[3] Haze Instruments d.o.o., Ljubljana, Slovenia
[4] University of Nova Gorica, Ajdovščina, Slovenia

**Abstract.** We have developed a novel single-beam photothermal interferometer and present here its application for the measurement of aerosol light absorption. The use of only a single laser beam allows for a compact optical set up and significantly easier alignment compared to standard dual-beam photothermal interferometers, making it ideal for field measurements. Due to a unique configuration of the reference interferometer arm, light absorption by aerosols can be determined directly even in the presence of light absorbing gases. The instrument can be calibrated directly with light absorbing gases, such as $NO_2$, and can be used to calibrate other light absorption instruments. The detection limits ($1\sigma$) for absorption for ten and sixty second averaging times were determined to be 14.6 Mm$^{-1}$ and 7.4 Mm$^{-1}$, respectively, which for a mass absorption cross-section of 10 m$^2$ g$^{-1}$ leads to equivalent black carbon concentration detection limits of 1460 ng m$^{-3}$ and 740 ng m$^{-3}$, respectively. The detection limit could be reduced further by improvements to the isolation of the instrument and the signal detection and processing schemes employed.

## 1. Introduction

According to estimates from the World Health Organization (WHO), particulate air pollution contributes to about seven million premature deaths each year, making it one of the leading causes of early mortality worldwide (WHO, 2014). Studies of short-term health effects suggest that black carbon (BC) particles, a component of carbonaceous aerosols, are a better indicator of the effect harmful particulate substances from combustion sources exert on human health than any other metric (Janssen et al., 2011; Janssen et al., 2012), and it is acknowledged that BC poses tremendous harm to public health. New estimates based on re-evaluation of data from across Europe suggest that air pollution leads to more than double the number of deaths than previously thought (Lelieveld et al., 2019).

Besides its health relevance, aerosol black carbon also significantly affects Earth's climate (Bond et al., 2013; IPCC, 2014). Aerosols influence our climate by their ability to scatter and absorb solar radiation (IPCC, 2014). For the assessment of the net aerosol direct radiative forcing, it is crucial that both, the aerosol light scattering as well as absorption properties, are measured accurately. As BC particles are highly efficient light absorbers, they are considered to be the second most important anthropogenic climate forcer after $CO_2$ (Bond et al., 2013). However, the uncertainty of the BC warming effect is still very high: the best estimate of the radiative forcing of BC is +1.1 W m$^{-2}$ (90% uncertainty bounds +0.17 to +2.1 W m$^{-2}$) (Bond et al., 2013). The magnitude of the warming effect strongly depends on the vertical placement in the atmosphere due to the reduction of solar radiation below the absorbing aerosols (Schwartz and Buseck, 2000; Penner et al., 2003; Streets et al., 2006), as well as the influence of aging on the aerosol optical properties (Zhang et al., 2018). It is therefore crucial to be able to measure aerosol absorption accurately in-situ.



Aerosol light absorption is quantified using the wavelength dependent absorption coefficient $b_{abs}(\lambda)$, which is defined as the attenuation of light due to absorption in the medium per unit length. The total attenuation of light passing through a sample is determined by the absorption and scattering ($b_{scat}$) coefficients using the Beer-Lambert law

$$I = I_0 e^{-(b_{abs}+b_{scat})\cdot x} \tag{1}$$

where $I$ is the intensity of light remaining after transmission through a medium of length $x$ given an initial intensity $I_0$. In order to relate aerosol light absorption to a mass concentration of (absorbing) aerosol particles the mass absorption cross-section (MAC) of the aerosol is required: $m = \frac{b_{abs}}{MAC}$, where $m$ is the mass concentration of the light absorbing aerosol component. For typical ambient BC aerosols measured at $\lambda = 532$ nm the MAC is approximately 10 m$^2$ g$^{-1}$ (extrapolated from measurements at 637 nm of 6.6 m$^2$ g$^{-1}$ (Petzold et al., 2002)), however

the uncertainty is this value is large due to the unavailability of traceable reference methods (Zanatta et al., 2016). In order to clarify the quantity that is measured in such experiments, Petzold et al. (2013) recommend to use the term equivalent black carbon (eBC) when its mass is derived by optical measurements.

Aerosol light absorption properties are commonly measured *ex-situ* using filter-based devices, such as the Aethalometer (Drinovec et al., 2015), Multi Angle Absorption Photometer (MAAP) (Petzold et al., 2002), Particle

Soot Absorption Photometer (PSAP) (Bond et al., 1999) and Continuous Light Absorption Photometer (CLAP) (Ogren et al., 2017). In such measurements, the aerosol particles are deposited onto a filter and the light transmission through the sample-laden filter is measured relative to the light transmission of the unloaded filter. The advantage of filter-based techniques is that they are straightforward, allow for unattended operation, and are relatively inexpensive. In addition, they have low detection limits due to the accumulation of the absorbing species

on the filter over time: the detection limits can reach $b_{abs} < 0.05$ Mm$^{-1}$ when the sample is collected over a sufficiently long time (Springston and Sedlacek, 2007; Backman et al., 2017). These methods have significant drawbacks however, as they suffer from large systematic errors caused by the modification of particle properties upon deposition in the filter (Weingartner et al., 2003; Lack et al., 2008; Drinovec et al., 2015; Drinovec et al., 2017). In addition, various optical interactions between the deposited particles and the filter medium can enhance

or lower the measured absorption. One major issue is the cross sensitivity to scattering material embedded in the filter, which enhances the apparent absorption (Arnott et al., 2005; Collaud Coen et al., 2010).

Due to the artefacts inherent in filter-based measurements, it is advantageous to measure the aerosol absorption with the particles suspended in the air. Several *in-situ* measurement techniques exist, amongst which the most common being the "extinction minus scattering" and photoacoustic methods. In the "extinction minus scattering"

method, light extinction and light scattering are measured separately, with light absorption defined as the difference between the measured quantities. The measurements can be very accurate, but encounter difficulties for aerosols featuring high single-scattering albedo $\frac{b_{scat}}{b_{scat}+b_{abs}}$ (above approx. 0.75), in which extinction and scattering are both large and almost equal quantities (Bond et al., 1999; Schnaiter et al., 2005). Unlike the "extinction minus scattering" method, instruments based on the photoacoustic effect measure the light absorption of the sample

directly as a pressure wave generated after the absorption of light in the aerosol and subsequent heating of the gas. As the photoacoustic signal is only generated by light absorption, artefacts from light scattering are completely eliminated. Photoacoustic measurements can be performed either non-resonantly or in an acoustic resonator, which amplifies the signal at the resonator mode frequencies (Tam, 1986). The photoacoustic method encounters a significant bias when measuring hygroscopic aerosols in elevated relative humidity (RH) or samples with volatile





coatings – this artefact arises from the loss of the latent heat of these particle-bound volatile species as they evaporate from the heated particles, reducing the apparent acoustic signal due to mass transfer (Arnott et al., 2003; Raspet et al., 2003; Murphy, 2009; Langridge et al., 2013). Recently, this artefact was studied for single micrometer-sized particles of tetraethylene glycol and a significant size dependence of the magnitude and sign of the artefact was observed (Diveky et al., 2019). Some photoacoustic instruments can achieve detection limits of

$b_{abs} \approx 0.1$ Mm$^{-1}$ (with 60 s averaging) (Lack et al., 2006), though most instruments have considerably higher detection limits (Linke et al., 2016).

*In situ* absorption methods have a further advantage over traditional measurements: the ability to traceably calibrate the response of the instrument using an absorbing gaseous species such as $NO_2$ (Arnott et al., 2000; Nakayama et al., 2015) or $O_3$ (Lack et al., 2006; Lack et al., 2012; Davies et al., 2018). Such internal primary

calibration standards are unavailable for filter-based instrumentation, which rely on comparative measurements with reference instruments and reference aerosols; not only are such calibration processes prone to biases, they also cannot be performed in the field, requiring the instrument to be shipped to the calibration facility.

Photothermal interferometry (PTI) is an *in-situ* direct absorption measurement technique originally developed for measurements of trace gases (Davis and Petuchowski, 1981; Fulghum and Tilleman, 1991; Mazzoni and Davis,

1991) that has also been considered for and applied to aerosol measurements (Davis and Petuchowski, 1981; Fluckiger et al., 1985; Lin and Campillo, 1985; Moosmüller and Arnott, 1996; Sedlacek, 2006; Sedlacek and Lee, 2007; Moosmüller et al., 2009; Lack et al., 2014; Lee and Moosmuller, 2020). In PTI, the light absorption $b_{abs}$ of a sample is measured by probing changes in the refractive index of the sample due to light absorption using interferometry. Previous realisations of PTI require two lasers, one of high power that is modulated and absorbed

by the sample (pump), and a second CW interferometry laser (probe). Upon absorption of pump beam light by the sample, energy is transferred via heat conduction to the buffer gas, which results in highly localised heating and thus a refractive index change within the pump beam volume. Light absorption is then proportional to the periodic phase change of the probe beam passing through the sample arm with respect to the reference arm. At the shot noise limit, the theoretical detection limit of PTI has been calculated to be $b_{abs} < 0.01$ Mm$^{-1}$ (30 s integration time)

(Sedlacek, 2006), though for aerosol measurements the practical detection limit is consistently considerably higher (e.g. $> 0.2$ Mm$^{-1}$ from Sedlacek and Lee (2007)).

The primary difficulties associated with achieving the theoretical detection limits are the sensitivity of interferometric measurements to external noise sources, the difficulty of optimally aligning and maintaining the alignment of the pump and probe beams, and measurement artefacts due to cross-sensitivity to other absorbing

species, such as $NO_2$, volatile organic compounds (VOC) and $O_3$. In order to tackle the problem of coupling external noise into the system, ever more complex isolation systems and interferometer configurations have been employed (Moosmüller and Arnott, 1996). Numerous pump-probe geometries have been explored to improve the detection limit. Geometries with better beam overlaps and thus higher measurement sensitivities typically also result in an increase in baseline noise due to the use of common optics, with no large improvement in the ease of

alignment (Sedlacek, 2006). The alignment of the laser beams, that is, the geometry of the pump-probe configuration, requires the instrument to be calibrated using a species of known absorption and this calibration must be repeated periodically to track the sensitivity (beam overlap) of the PTI. Otherwise, uncorrected changes in sensitivity are interpreted as changes of the aerosol light absorption. Small changes in lasing mode or beam shape in either laser can alter the sensitivity of the measurement enough to significantly affect measurement



accuracy. Measurement artefacts due to the cross sensitivity of PTI to other absorbing species have typically been compensated for by simultaneously measuring a filtered sample stream and subtracting the offset.

In order to address the aforementioned difficulties of the PTI technique, a new PTI instrument employing a single laser and unique beam configuration has been developed. This instrument greatly simplifies the alignment of the interferometer, maximises the sensitivity of the measurement and enables artefact free measurement of aerosol

absorption in the presence of absorbing gaseous species. We report here on the experimental realisation of this instrument, which we have termed the modulated single-beam PTI (MSPTI) configuration, its initial characterisation with $NO_2$ and first laboratory measurements of carbonaceous aerosols.

## 2. Standard pump-probe PTI

In the PTI technique, the absorption based induced change of the refractive index of the buffer gas is detected via

the relative phase shift of light waves passing through the sample versus those traversing the reference medium within the interferometer. Substances (particles, molecules, etc.) that absorb light at the pump laser wavelength transfer the absorbed energy to the surrounding buffer gas via heat conduction, resulting in a local increase of the buffer gas temperature. As the refractive index of a gas depends upon its density, which is itself temperature dependent, modulation of the pump laser intensity in the presence of a light absorbing substance results in the local

modulation of the refractive index. Light passing through this volume experiences a periodic phase shift, which can be measured via interferometry. Figure 1 shows a schematic of this process.

The measured phase shift $\Delta\varphi$ is related the absorption coefficient $b_{abs}$ via the following relation (Moosmüller et al., 1997; Sedlacek, 2006):

$$\Delta\varphi = \frac{2\pi(n-1)}{\lambda_{Probe}T\rho C_p}\frac{lP_{Pump}}{A}b_{abs}\Delta t \qquad (2)$$

The first term can be considered constant for a given temperature, where $n$, $T$, $\rho$ and $C_p$ are the refractive index, temperature, density and heat capacity of the air, respectively. $\lambda_{Probe}$ is the interferometer laser wavelength and $\Delta t$ denotes the heating time within the modulation cycle. The second term can be defined as the sensitivity of the

PTI measurement, where $l$ is the length of the overlap of the beams within the sample volume, $P_{Pump}$ is the modulation amplitude of the pump beam power and $A$ is the effective cross-sectional area of the laser beams. Therefore, maximum sensitivity is achieved by maximising the length of the interaction and the pump laser power and minimising the cross-sectional area of the beams (though requiring that the cross sections overlap).

In standard realisations of PTI, a single sample chamber is placed either in the measurement arm alone or across

both arms of the interferometer. One potential realisation of a standard PTI set up is shown in Figure 2. The modulated pump beam is set to overlap the probe beam within the sample chamber in the measurement (lower) arm. As interferometric measurements are relative, the resultant signal contains an AC (modulated) component associated with light absorption and a DC offset due to an optical path length difference between the two arms of the interferometer. It is important to note in this case that the AC signal results from the total light absorption of

the sample, regardless of the absorbing species, that is, it includes absorption in gases and aerosol in the air sample carried through the aerosol chamber.

Interferometric detection schemes operate by comparing the phase of two light waves, which have traversed different paths and interfere at the detection point. The two different beam paths through the interferometer are



typically referred to as arms; one arm is isolated from the environment and designated the reference arm, while
the other is employed for measurement purposes. After traversing the respective arms of the interferometer, the
light waves are recombined and split anew into two separate output beams at a beam splitter, with each output
beam ideally containing 50% of the power from the laser beams traversing the reference and measurement arms,
respectively. The light waves in the output beams interfere, causing the measured intensities of the output beams
to vary with the phase difference between the measurement and reference light waves. The light intensity in each
output beam (measured with the respective detectors) can then be defined in terms of this phase difference
$\varphi_{meas}$ - $\varphi_{ref}$, which we will assign as the interferometric phase $\varphi_{inf}$ and is given by

$$I_1 = I_0 \sin^2\left(\frac{\varphi_{inf}}{2}\right) = \frac{1}{2}I_0\left(1 + \cos(\varphi_{inf})\right) \tag{3}$$

$$I_2 = I_0 \cos^2\left(\frac{\varphi_{inf}}{2}\right) = \frac{1}{2}I_0\left(1 - \cos(\varphi_{inf})\right) \tag{4}$$

where $I_0$ is the intensity of the laser before the initial beam-splitter.

The complex amplitude of the combined beams is the sum of the individual beam amplitudes – they interfere. This
leads to the limits of totally constructive $I_x = I_0$ and totally destructive interference $I_x = 0$ at each detector. These
limits occur for phase differences of a multiple of $\pi$. For example, constructive interference occurs at D$_1$ for a
phase difference of $2n\pi$, $n$ = 0, 1, 2…, and destructive interference for a phase difference of $(2n+1)\pi$, $n$ = 0, 1,
2…. This relationship is shifted by $\pi$ radians for D$_2$, thus maintaining conservation of energy. The relationship
between the intensity measured at the detectors and the phase difference between the interfering light waves for
an ideal interferometer is shown in Figure 3.

For the measurement of small time-dependent phase shift $\Delta\varphi(t)$, such as that produced via light absorption in PTI,
it is necessary to consider the interferometric phase $\varphi_{inf}$. Due to the sinusoidal nature of the measured signals, the
relationship between a small change in the phase difference between the waves $\Delta\varphi(t)$ and the change of the
measured intensity $\Delta I$ is not constant and depends on $\varphi_{inf}$. This property of the measurement is shown in Figure
3. At $\varphi_{inf} = (2n + 1)\frac{\pi}{2}$, where n = 0, 1, 2…, the slope $\left|\frac{\Delta I}{\Delta\varphi}\right|$ and thus the sensitivity of the measurement is
maximised. In order to take advantage of this, PTI measurements are typically performed in phase quadrature by
actively regulating $\varphi_{inf}$ to $(2n + 1)\frac{\pi}{2}$. Previous PTI instruments (Moosmüller and Arnott, 1996; Sedlacek, 2006;
Sedlacek and Lee, 2007) have used a number of different methods to regulate the phase difference between the
two arms of the interferometer, with the application of a piezo-electric element to move one of the interferometer
optical elements being the most common solution.

At the quadrature points the relationship between a sufficiently small phase change $\Delta\varphi(t)$ (e.g. for $\sin \Delta\varphi \cong \Delta\varphi$)
and the measured intensities can be approximated by:

$$\Delta\varphi(t) = \left|\frac{I_1 - I_2}{I_1 + I_2}\right| \tag{5}$$

where $I_1$ and $I_2$ are the intensities of light measured by the detectors in the two outputs of the interferometer as a
function of time. In the ideal case of a light source of constant intensity and an optically thin medium, $I_1 + I_2$ is
constant.

Phase shifts may additionally arise from sources other than the photothermal effect, such as from acoustic noise
and changes of the length of the interferometer due to vibrations. Low frequency noise can be separated from
phase shifts due to the photothermal effect by modulating the pump laser at a higher frequency and restricting the
detection bandwidth to this frequency. This is typically performed experimentally using a lock-in amplifier.



Unwanted variations in phase (phase noise) can be reduced through the choice of the interferometer geometry. The current preferred interferometer geometry for aerosol measurements is a folded one, for example a folded Jamin (Moosmüller and Arnott, 1996; Sedlacek, 2006). This design minimises the influence of interferometric noise by

placing both arms of the interferometer in parallel and close proximity and through the use of an etalon and retroreflector. The use of these two optical components ensures that any external noise that is coupled into the interferometer affects both arms equally and thus cancels out.

### 3. Modulated single-beam PTI

Here we present a new PTI configuration, which we have named the MSPTI, in which the pump and probe beams

are replaced by a single modulated laser beam. This beam has the same optical path as the probe beam in a conventional PTI set up and is modulated between two sufficiently different intensity levels (i.e. $I_{high} > 10 \cdot I_{low}$). The major advantage of the modulated configuration is the simplified optical alignment of the system. As a single beam fulfils both pump and probe functions, the pump-probe co-incident volume is, by definition, the entire beam volume. This ensures maximum heating and detection sensitivities and enables the use of significantly lower pump

intensities, thus reducing the potential for the destruction of the sample and measurement noise arising from heating of the interferometer optical components.

The MSPTI configuration requires a different approach to signal evaluation than standard PTI. As the probe beam is not maintained at constant intensity, additional data analysis steps are required in order to optimally extract the PTI signal encoded on the modulated laser beam. The MSPTI configuration also places additional constraints on

the single laser employed. The standard requirements for the interferometric probe beam of low noise and significant coherence length remain, but high CW power is additionally required. For the MSPTI prototype, we have chosen to employ a diode pumped solid state laser operating at 532 nm, modulated with an external acousto-optic modulator to improve rise and fall times as well as laser stability.

The measured phase shift in the MSPTI is equivalent to the two-beam case, with only slight modification required

to the formulae. The pump and probe subscripts are dropped from Equation 2 and $P_{Pump}$ is replaced by $\Delta P$, the modulated laser power in the measurement chamber (for the case of a 50:50 beam splitter and no optical losses $\Delta P$ is half of the modulated power exiting the laser). Performing these substitutions results in Equation 6, which is valid for the simplified case of a laser beam with constant diameter.

$$\Delta\varphi = \frac{2\pi(n-1)}{\lambda T \rho C_p} \frac{l\Delta P}{A} b_{abs} \Delta t \tag{6}$$

For the case of a focused laser beam with a Gaussian intensity distribution with focal point in the middle of measurement chamber, it can be shown (see the Supplementary Information) that the phase change due to the PTI effect is:

$$\Delta\varphi = \frac{2\pi(n-1)}{\lambda T \rho C_p} \frac{2\Delta P}{\lambda} \tan^{-1}\left(\frac{a}{z_r}\right) b_{abs} \Delta t \tag{7}$$

where $2a$ is the length of the measurement chamber and $z_r$ is the Rayleigh distance for the modulated beam

focused in the middle of the measurement chamber. For the case where the length of the chamber is twice the Rayleigh distance $a = z_r$, and the sensitivity becomes $\cong 0.79 \cdot \frac{2\Delta P}{\lambda}$. The sensitivity of the measurement approaches the limit $\tan^{-1}\left(\frac{a}{z_r}\right) \approx \frac{\pi}{2} \cdot \frac{2\Delta P}{\lambda}$, for $a > 2 \cdot z_r$. Thus, the maximum sensitivity achievable for an arbitrary interaction (chamber) length is limited and cannot be further increased through improved focusing.



Unlike for the case of standard PTI, in which the small phase changes are measured from a stable CW laser
intensity, the MSPTI signal is dominated by the modulation of the laser intensity. Thus, lock-in detection cannot
be directly performed with the difference signal of the interferometer outputs and a normalisation step is required.
This is performed by normalising the time-dependent difference signal from the detectors by the total light intensity
(refer to Equation 3 above). This step accounts for the dependence of the phase change signal on the total light
intensity of the interferometer beam. An example of this normalisation step is shown in Figure 4.

Thus, a new PTI signal processing method was developed to address the additional complications of the MSPTI
method, when compared to standard pump-probe PTI. Signal processing was performed in software, after
digitising the raw signals from the photodetectors. As the desired quantity from the measurement is the magnitude
of the light absorption by the aerosol, it was sufficient to analyse the heating (or high) phases alone. To ensure
maximum signal-to-noise ratio and avoid reducing the data further, the heating curves were analysed in full. An
example heating curve calculated from Eq. 5 for approximately 100 µg m$^{-3}$ of electrical discharge soot is shown
in Figure 5. From Eq. 6 it could be expected that the phase shift due to light absorption should increase linearly
with the duration of the heating phase $\Delta t$. However, the example heating curve shown in Figure 5 deviates
considerably from linearity as the heating phase progresses. This is due to the loss of absorbed energy in the form
of heat out of the sensing (laser) volume with time. Taken to the limit of $\Delta t \rightarrow \infty$ for a non-modulated laser beam,
Eq. 6 implies that $\Delta\varphi \rightarrow \infty$, however in reality, $\Delta\varphi$ approaches equilibrium as the heat arising from absorption of
the laser beam is balanced by the heat flowing out of the detection volume. Thus, Equation 6 is only valid for
heating times $\Delta t$ below a characteristic value.

Empirically, the best fit to the data was found to be an exponential of the form:

$$\Delta\varphi(t_{heat}) = a\left(1 - e^{-\frac{t}{\tau}}\right) + c \tag{8}$$

where $a$ is a parameter representing the limit of the phase change due to the temperature increase of the sample
volume due to light absorption and temperature loss outside of the laser beam volume, $\tau$ is the mean lifetime of
the cooling process and is dependent on the beam geometry and $c$ is the absolute offset from phase quadrature.
Equation 8 is closely related to Newton's Law of Cooling adapted to be expressed in terms of phase shift and with
the addition of a heating term due to light absorption during the heating cycle. An example of the least squares
best fit of this form is shown as the dashed line in Figure 5.

As the characteristic cooling time $\tau$ is predominantly dependent on the geometry of the heating/sensing volume
and this does not change during measurements, further simplification of the fit to obtain $\Delta\varphi(t_{heat})$ is possible. It
was found that for a specific range of heating times that the exponential fit could be approximated with a linear
one, with slope $\frac{d\Delta\varphi(t_{heat})}{dt}$. An example least squares linear fit to a heating curve is shown as the dotted line in Figure
5. The quality of the linear fit to the data appears poor; however, it still contains the required information for the
calculation of $\Delta\varphi(t_{heat})$ and $b_{abs}$ when calibrated using a species of known absorption. The interested reader is
directed to the Supplementary Information for additional details.

It should be noted that this analysis only holds for the specific cases of a single-beam modulated interferometer
and a two-beam interferometer with exactly equal perfectly overlapped pump and probe volumes. For the general
case of significantly different pump and probe beam geometries, the dynamics of the system will differ
considerably from those obtained in this work (Monson et al., 1989).



## 4. Modulated single-beam PTI experimental setup

The physical layout of the interferometer is based on the folded Jamin interferometers of Moosmüller and Sedlacek (Moosmüller and Arnott, 1996; Sedlacek, 2006) and is shown in Figure 6. The etalon in the Jamin design has been

replaced by separate beam splitter and mirror optics, which are mounted in a solid metal block. The overlap of the interfering beams can be adjusted with the positioning of the mirror by way of thumbscrews. The resulting layout is a folded Mach-Zehnder interferometer. As both the reference and measurement beams are incident on the beam splitter and mirror, the effects of solid borne noise are reduced when compared to standard Michelson or Mach-Zehnder designs. The insensitivity to external noise is not as complete as for the Jamin design, as the two optical

elements are able to move with respect to each other, but the design does allow for flexibility in the design of the aerosol chamber.

The MSPTI design additionally requires a different aerosol chamber design compared to standard PTI instruments. In the case of MSPTI, the single modulated beam is present in both reference and measurement arms of the interferometer and therefore a difference in aerosol composition between the reference and measurement arms is

required. In the current MSPTI prototype the aerosol chamber consists of three isolated cells, one for the measurement arm (sample) and two for the reference arm. An absolute filter separates the sample and reference cells. A schematic of the flow set up for the MSPTI instrument is shown in Figure 7.

The effective perfect beam overlap for the MSPTI in both sample and reference chambers confers an additional advantage – the ability to directly subtract absorption by gaseous species. As the light absorbing gaseous species

are present in the same concentration in both arms of the interferometer, the photothermal effect due to these gaseous species is the same and the net phase difference is zero. This compensation of the gas absorption requires both equal laser intensity in the sample and reference arms of the interferometer and equal sensitivity due to the positioning of the two focal points. Equal intensity can be achieved with a 50:50 beam splitter (at the laser wavelength). Equal sensitivities also require that the Rayleigh distance from the focal points lie entirely within the

respective chambers. Fulfilment of these prerequisites thus enables the determination of the light absorption of the aerosol only, even though the complex sample mixture may additionally feature absorbing gases and light scattering aerosol.

In the MSPTI instrument, phase quadrature is actively regulated using a pressure cell in the reference beam path. As the refractive index of a gas depends on pressure, the optical path length of the reference path can be adjusted

by varying the pressure in the cell (positioned between the reference arm and the retroreflector in Figure 6). The quadrature regulation is performed at frequencies below 1 Hz in order to counteract slow changes in the optical path lengths, such as from thermal drifts or changes in refractive index of the gas. The advantage of this method is its simplicity (does not require the production of custom optics) and the lack of moving parts.

The alignment of the interferometer is comparatively simple. The modulated beam is coupled into the

interferometer block at an angle of 45 degrees relative to the beam splitter, after which the retroreflector is adjusted so that the returning beams pass through the cells in the aerosol chamber as required. The focusing lens is then inserted before the interferometer block and its position adjusted so that the focus is centred within the aerosol chamber. Finally, the overlap of the interfering beams is optimised by adjusting the mirror in the interferometer block until it is parallel with the beam splitter and maximum interferometric contrast is acquired. No further

adjustment of beam overlap is required.



The PTI instrument is mounted on an optical breadboard (Thorlabs, B60120A). Solid borne vibrations are damped using a set of passive vibration isolators (Thorlabs, PWA090). The interferometer is housed within a metal box lined with acoustic foam for isolation from air currents and external noise sources.

The laser source is a diode pumped solid-state laser (Laser Quantum, GEM 450 mW) at 532 nm. Before entering the interferometer the beam first passes through a half-wave plate (Thorlabs, WPH05M-532), which rotates the polarisation to vertical. Intensity modulation of the beam is performed with an acousto-optic modulator (AOM) (AA Optoelectronic, AA.MT110-A1,5-VIS) and the $0^{th}$ order output is selected in order to maximise the available laser power for PTI. Subsequently, the beam is expanded by a factor of three by a Galilean beam expander and the polarisation rotated by 45° for optimal splitting at the non-polarising beam splitter in the interferometer. All mirrors employed in the interferometer are broadband dielectric mirrors designed for use at visible wavelengths (Thorlabs, BB1-E02).

The modulated interferometer in this work is of folded Mach-Zehnder design and consists of a broadband dielectric mirror (Thorlabs, BBSQ2-E02), a 50:50 amplitude splitting beam splitter optimised for 532 nm (Thorlabs, BSW4R-532) and a 50.8 mm diameter retroreflector (Edmund Optics, #49-666). The mirror and beam splitter are mounted into a single custom machined metal block in order to reduce relative movements of the optics and shift solid borne vibrations to higher frequencies. The mirror tilt is adjustable in two planes via thumbscrews in order to align the beams and achieve maximum interferometric contrast (97% typical). The laser focus inside the interferometer was checked using a CCD camera (Basler, acA1300-30um) and optimised by adjusting the position of the focusing lens (Thorlabs, LA1908-A).

Custom-built aerosol and pressure chambers are situated within the interferometer. Each of the chambers consists of three individual cells, which are separately sealed using optical windows (Edmund Optics, #46-100) with O-rings. The reference arm of the interferometer consists of the two outer cells of each chamber. The central sample cell comprises the measurement arm, through which the laser passes twice. Samples are introduced into the central cell and then are either exhausted ($NO_2$ calibration measurements) or flow through an absolute filter and then each outer cell in series (standard aerosol measurements) as shown in Figure 5. The gas flow is subsequently measured using a flowmeter (WISAG, 1000 series). One outer cell of the pressure cell is connected to a regulated pressure valve (Parker, 980-005101-015) and is controlled by a software based proportional-integral (PI) controller.

In the detection component, the diverging beams are refocused using biconvex lenses (Thorlabs, LB1945-A-ML) and the optical power reduced with ND filters (Thorlabs, NE10A). Detection of the interfering laser beams is performed with a photodiode in each interferometer output (Thorlabs, DET36A) operating in photoconductive mode. The use of two detectors allows the rejection of false signals, such as changes in laser intensity. The detected photocurrents are converted into voltages using a 1.2 k$\Omega$ resistor in parallel to the photodiode and subsequently digitised (National Instruments, NI USB-6356).

Carbonaceous aerosols are generated with a spark discharge soot generator (PALAS, GFG 1000). Argon (Messer, 4.8) is used as the inert carrier gas for the discharge and subsequent transport of the generated particles. Comparison measurements of eBC concentrations are performed using an Aethalometer (Aerosol d.o.o., AE33). $NO_2$ concentrations are prepared from a mixture of 1 ppm NO2 in synthetic air (Messer) in excess synthetic air (Messer, 5.6) using mass flow controllers (Voegtlin Instruments, red-y GSC-B9SA-DD23 and –DD26) and a $NO_2$ monitor (Horiba, APNA-370) was available for reference concentration measurements.

## 4. Results


PTI is an *in situ* light absorption measurement technique and as such, it is possible to use an absorbing gas to calibrate the sensitivity of the instrument. In the visible range, $NO_2$ and $O_3$ gases have the highest absorption cross-sections and $NO_2$ was chosen as the calibration gas for this study. The initial characterisation of the MSPTI was additionally performed with $NO_2$, in order to determine the optimal operating conditions for the instrument. The

optimal operating frequency can be determined by investigating the relationship between the duration of the laser high period (heating time) and the resultant PTI signal. As can be seen in Equation 6, the measured phase shift for PTI is linearly dependent on the heating time. This however, is only valid for a specific range of heating times. If the heating time exceeds a characteristic value $\tau > \frac{w^2}{D}$, dependent on the beam radius $w$ and the gas thermal diffusivity $D$, then this equality is no longer maintained as heat flows out of the sensing volume during the

measurement and no longer contributes to the signal (Monson et al., 1989). This is observed as a flattening of the heating curves with increasing heating time and results in the apparent reduction of absorption. The calculations of Monson *et al.* were performed for collimated laser beams and under the condition that the probe beam diameter was much smaller than that of the pump beam. In the current case, the pump and probe diameters are equal and vary with $z$, leading to a non-cylindrically symmetric temperature distribution along $z$. If we however assume an

average beam radius of 0.1 mm within the sample cell, then $\frac{w^2}{D}$ becomes 0.53 ms for a gas thermal diffusivity of 19 $mm^2$ $s^{-1}$. This is in reasonable agreement with the heating curve shown in Figure 5, in which the deviation of the observed phase shift from linearity begins at ∼ 0.7 ms.

In order to determine the optimal modulation frequency of the laser beam, the PTI signal for 1 ppm $NO_2$ was measured for a range of different modulation frequencies. The results of these measurements are shown in Figure

8 as a function of heating time. In order to enable comparison with previous works, the measured signals have been converted to phase shifts with units of radians. At shorter heating times the PTI signal is observed to increase linearly as a function of the heating time, in agreement with Equation 6. The transition out of the linear regime occurs at heating time of ∼5 ms (∼100 Hz modulation frequency). This result is consistent with the value previously reported by Sedlacek (Sedlacek, 2006) for a two beam experimental PTI set up. It must be noted here

that the existence of a linear relationship between the PTI signal and heating time does not imply that the heating curves themselves increase linearly with time. In fact, close examination of Figure 5 shows that this is very clearly not the case for the MSPTI. Instead the linear relationship between PTI signal and heating time implies that the shape of the heating curves remain similar for this range of heating times, the chosen evaluation of the PTI signal (e.g. linear fit or lock-in detection) is in good agreement with Eq. 6 and a calibration performed at one heating

time can be directly transferred to measurements performed at a different heating time.

A modulation frequency of 91 Hz was chosen in this study in order to avoid a significant noise band around 100 Hz, which was observed during the initial laboratory measurements. Operation at 91 Hz ensured a larger signal than operation at frequencies above 100 Hz, but meant that the MSPTI was operated outside of the afore-described linear regime. Therefore, calibration measurements were also performed at 91 Hz such that the application of Eq.

6 was valid for this modulation frequency. This is explained further in the following paragraphs detailing the calibration procedure employed in this work.

The sensitivity of the instrument was experimentally determined from the MSPTI signal dependence on $NO_2$ concentration. For these measurements, the aerosol chamber was connected so that the $NO_2$ flowed through the sample chamber and then was exhausted (see Figure 7a, calibration configuration). The reference chambers were



filled with synthetic air for the calibration procedure. Measurements were performed for 0.2 to 1 ppm of $NO_2$ in synthetic air at a flow rate of 0.5 l min$^{-1}$ and the results are plotted in Figure 9. The two data sets represent two separate measurements, where the concentration of $NO_2$ was firstly increased stepwise from 0 ppm to 1 ppm and then decreased back to 0 ppm. No obvious measurement hysteresis was observed between the data sets. The offset of the measurement is attributed to PTI signals generated by the optical components in the interferometer.

The data show a clear linear relationship between the PTI signal and $NO_2$ mixing concentration set in the flow system and show the validity of the developed signal analysis. From the slope of the concentration curve and the absorption cross-section of $NO_2$ reported in literature the sensitivity ($\frac{l\Delta P}{A}$) for the MSPTI instrument can be calculated with Equation 6. The absorption cross-section of $NO_2$ used was 1.47x10$^{-19}$ cm$^2$ molecule$^{-1}$, which was obtained by convoluting the data of Vandaele et al. (2002) (accessed from the MPI-Mainz UV-VIS Spectral Atlas

(Keller-Rudek et al., 2013)) with a Gaussian function at the reported laser wavelength of 532.075 nm and spectral bandwidth of 30 GHz. Using this value and typical literature values for air at standard temperature and pressure, a sensitivity ($\frac{l\Delta P}{A}$) of 6.80 kW m$^{-1}$ is calculated for the MSPTI system. This value is approximately two orders of magnitude lower than the theoretical value determined by measuring the properties of the laser beam at its focus (see Supplementary Information for the calculation). One major reason for this is the application of a linear fit to

the heating curves in this study (see Figure 5 and accompanying text), which is significantly affected by heat loss out of the laser volume during the measurement. As Equations 6 and 7 do not account for the heat loss out of the measurement volume during the measurement, any signal loss due to this process negatively impacts the magnitude of $\frac{l\Delta P}{A}$ determined from the calibration measurements. This is true for measurements performed at modulation frequencies both within and outside the linear range of Figure 8, but the effect is larger for modulation frequencies

outside of the linear range. The reason for this can be seen in the linear fit of Figure 5. If the deviation from linearity of the PTI signal with heating time due to heat loss out of the measurement volume could be excluded from the measurement, then both the measured signal and, by extension, the sensitivity $\frac{l\Delta P}{A}$ determined from calibration measurements would be considerably higher. An exponential fit to the heating curves and subsequent use of $\frac{a}{\tau}$ for the determination of the phase change due to light absorption would lead the magnitude of the sensitivity determined from the calibration to approach the value obtained by measuring the beam parameters.

Besides the ability to perform a primary calibration of the MSPTI with $NO_2$, a further advantage of the instrumental design over existing PTI instruments is the ability to directly differentiate aerosol absorption from absorption from gaseous species during a standard measurement. This removes the need to intermittently determine the absorption background from gases using filtered measurements as per other techniques. In the standard measurement

configuration, aerosol enters the sample chamber, passes an absolute filter and then flows through the reference chamber. The absolute filter traps the aerosol particles, but transmits the gas, which then flows through the reference chambers. To demonstrate the relative nature of PTI measurements and the advantages of the MSPTI set up in separating gas and aerosol absorption, comparison measurements of $NO_2$ were performed using the calibration and standard flow set ups. The results of these measurements for a flow rate of 0.5 l min$^{-1}$ are presented

in Figure 10. For the calibration flow set up, the $NO_2$ is only present in the sample cell and a PTI signal is measured. In the standard flow set up, $NO_2$ is present at the same concentration in the sample and reference cells and no signal is observed within experimental error. Thus, with the new MSPTI configuration, aerosol absorption can be measured independent of gas absorption, which reduces possible artefacts in the determination of aerosol absorption in ambient measurements. This is a significant advantage over previous PTI designs, which rely on



either periodic measurements of the background gas absorption or the measurements of other sensors to determine the aerosol absorption from the total absorption.

The detection limit dependence on the averaging time was determined from a baseline measurement with no flow, during which no effort was made to control noise in the laboratory. The raw data from the baseline measurement show a linear drift in apparent absorption most probably due to a slow change in splitting ratio of the beam splitter

over time, causing a change in the laser power in both arms of the interferometer and therefore the absorption by optical elements in the each arm. The exact source of the drift is yet to be confirmed as it is complicated by positive and negative contributions by optics in either interferometer arm. The drift, however, could be easily corrected by subtracting a linear fit from the data. The data and linear fit for evaluation of the drift are shown in the Supporting Information. In standard measurements, the baseline is determined at regular intervals with filtered air to account

for such baseline drifts. The experimentally measured standard deviations for the raw and drift corrected data are shown in Figure 11. The standard deviation of the raw data is dominated by the observed drift and improves very little for longer averaging durations. The standard deviation of the drift corrected data reduces at a rate of approximately $t^{-\frac{1}{2}}$ up to an averaging time of 60 seconds, after which the rate of reduction reduces. For the calculation of the detection limits of the instrument ($1\sigma$), standard deviations for averaging times of 1, 10, 60 and

300 seconds were compared to the $NO_2$ calibration measurement in Figure 9. Baseline drift corrected detection limits of our current instrument are summarized in terms of MSPTI signal (rad $s^{-1}$), $b_{abs}$, $NO_2$ concentration and eBC for selected integration times in Table 1.

Initial measurements of aerosols were performed by sampling from a reservoir pre-filled with graphitic soot produced using a spark discharge source and diluted with laboratory air. The aerosol was sampled from the volume

at a rate of 0.25 l min$^{-1}$ through the PTI in the measurement flow configuration using a pump connected to the outlet of the interferometer flow system. Comparison measurements were made using an Aethalometer AE33 sampling at 2 l min$^{-1}$, which was connected separately to the sampling volume in order to reduce the contribution from the AE33 pump to the MSPTI signal noise. For the AE33, the loading corrected eBC values (BC3) measured at $\lambda = 520$ nm were converted into absorption coefficients according to

$$b_{abs} = \text{eBC} \cdot \text{MAC}_{instr} \cdot \frac{C_{instr}}{C_{new}} \qquad (9)$$

where $\text{MAC}_{instr} = 13.14$ m$^2$ g$^{-1}$ and $C_{instr} = 1.57$ are the MAC and filter multiple-scattering enhancement parameter values used by the instruments' firmware to report eBC values, respectively. This conversion was corrected with the updated value of $C_{new}=2.6$ (value from (WMO, 2016), normalized as in (Drinovec et al., 2015)).

The MSPTI signal was converted to $b_{abs}$ using the sensitivity calculated from the $NO_2$ calibration (*vide supra*). The results of a typical measurement are presented in Figure 12. The AE33 signal shows strong loading artefacts at high soot concentrations, which are not completely corrected by the internal correction algorithm (Drinovec et al., 2015). Also seen are the automated filter changes of the AE33, indicated by black arrows, which are triggered when the light extinction through the sample spot exceeds a pre-specified value. A discontinuity can be seen at

around 15:15 in the MSPTI data, indicated by a red arrow, which is assigned to a laser mode hop. This leads to a change in the background level and as such can be corrected using regular background measurements. This artefact has been left uncorrected in order to show the effects of laser noise in the measurement. Improvements to the laser cooling system will be made in future development of the interferometer in order to stabilise the temperature and thus lasing mode of the DPSS laser.

The noise visible in the MSPTI signal can be attributed to several sources. Large outliers in the MSPTI signal are attributed to solid borne shocks that are transported to the interferometer. Other obvious outliers can be assigned to imperfect isolation of the interferometer to external acoustic noise sources (pumps, discussions in the laboratory) and laser instability. Additionally, noise from the pump in the AE33 is coupled through the sampling reservoir and adds to the baseline noise of the PTI measurement. This can be seen at 17:30 in the measurement data, when an

absolute filter was inserted into the sampling line to determine the background signal level, thus better isolating the MSPTI from noise transported from the sampling volume through the sampling line. The standard deviation of the data collected with the filter is a factor of 6.5 lower than the aerosol measurement and shows the need for an improvement in decoupling the MSPTI from external noise sources via the sampling line. For this measurement with the associated noise sources, the MSPTI instrument was determined to have a higher detection limit ($1\sigma$) of

approximately $b_{abs} = 17$ Mm$^{-1}$ ($\sim$1.7 µg m$^{-3}$) for an averaging period of 10 seconds, which agrees well with the values determined for the measurements of $NO_2$ summarised in Table 1.

Future improvements to the MSPTI set up are primarily targeted at the reduction of noise in the measurements. The laboratory tests show that an improvement in the isolation of the interferometer is required when operating in noisy environments. New outer and inner enclosures for the MSPTI are currently being evaluated for this purpose.

Improvements to the data analysis system to reduce the detection bandwidth and thus noise rejection are ongoing and are expected to bring a significant improvement in the detection limit. First measurements with 400 mW (200 mW per interferometer arm) laser power show a two-fold signal increase with no associated increase in noise. This implies that the system is interferometer noise limited and a further increase of the laser power would lead to an equal improvement in the detection limit. Potential improvement is foreseen by employing a balanced photodiode

detector and amplifying the difference signal to better employ the full range of the ADC.

## 5. Conclusion

We have demonstrated a new PTI prototype utilising only a single laser with a significantly improved ease of alignment compared to existing PTI instruments. The MSPTI design also allows for the direct measurement of aerosol absorption in the presence of absorbing gases, which would normally require a complicated correction or

secondary measurement of the gas absorption for other *in-situ* aerosol absorption measurements.

With a detection limit of in the range of a couple µg m$^{-3}$ of eBC for an integration time of 10 seconds the MSPTI set up does not currently improve upon the best reported detection limits for PTI measurements of aerosols (see e.g. (Sedlacek and Lee, 2007)), but simplifies its operational use in the field. Improvements to the isolation of the interferometer and data handling and analysis are expected to reduce the detection limit to the point where

unattended field measurements of ambient aerosol concentrations are possible.

The improvement in handling and alignment of the interferometer is significant and of great advantage when operated by non-experts in field measurements. Furthermore, the ability to directly measure aerosol absorption without bias from light absorbing gaseous species further reduces the potential for measurement artefacts due to concentration changes of these species. It also opens up the potential to employ the MSPTI in emission

measurements, where the concentrations of absorbing gaseous species can be significant and fluctuate rapidly.



**Author contributions**

BV, JR, PS and EW designed and developed the MSPTI. BV and JR carried out the experiments and analysed the data. LD and GM provided input into experimental design. BV wrote the manuscript, with JR, EW, LD and GM providing valuable additions.

**Competing interests**

The authors declare that they have no conflict of interest. LD and GM are employed at the potential future manufacturer of the instrument.

**Acknowledgements**

The research leading to these results has received funding from the Swiss National Science Foundation (SNSF)
grant no. 200021_172649, the 16ENV02 Black Carbon project of the European Union through the European Metrology Programme for Innovation and Research (EMPIR), and the EUROSTARS grant E!11386 IMALA. We thank J. Lee for his help and encouragement in the initial stages of the project, M. Gysel (PSI) for the use of the AE33, C. Hüglin (EMPA) for the loan of the $NO_2$ Monitor, A. Sedlacek, M. Krejci and H. Looser for fruitful discussions, S. Sjögren for the initial work on evaluating PTI configurations and A. Meier, M. Wipf, D. Egli, P.
Specht for their assistance and technical expertise. We thank AMES d.o.o. for technical support.

A patent application was filed on the described technology.

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


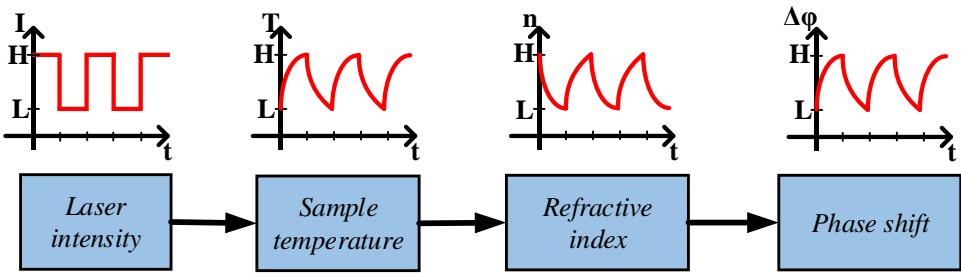


**Figure 1 – Scheme of PTI signal generation and measurement. For standard pump-probe PTI measurements the laser power at the low level L = 0.**

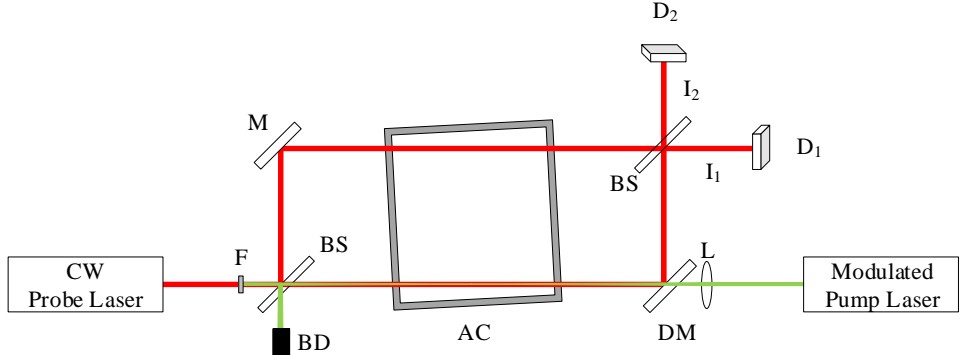

**Figure 2 – A potential realisation of a standard two-beam photothermal interferometer, similar to published configurations (see e.g. (Lee and Moosmuller, 2020)). The pump laser is set to overlap the probe beam in the measurement arm of the interferometer within the aerosol chamber. BS are beam splitters, M a mirror, DM a dichroic mirror, F a band pass filter, BD a beam dump, AC an aerosol chamber and D are detectors.**

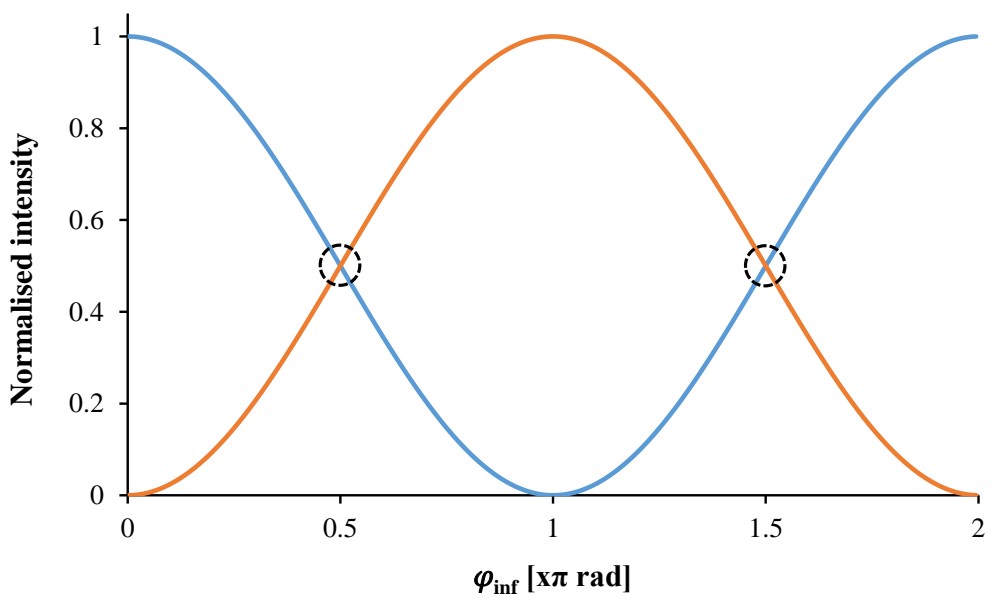


**Figure 3 – The intensity of light measured by each detector in the interferometer as a function of the interferometric phase. The dashed circles indicate the quadrature points, where the light intensity falling on each detector is approximately equal. At these points the relationship between a small phase shift $\Delta\varphi$ and the measured intensities are approximately linear and the sensitivity of the measurement is maximal.**


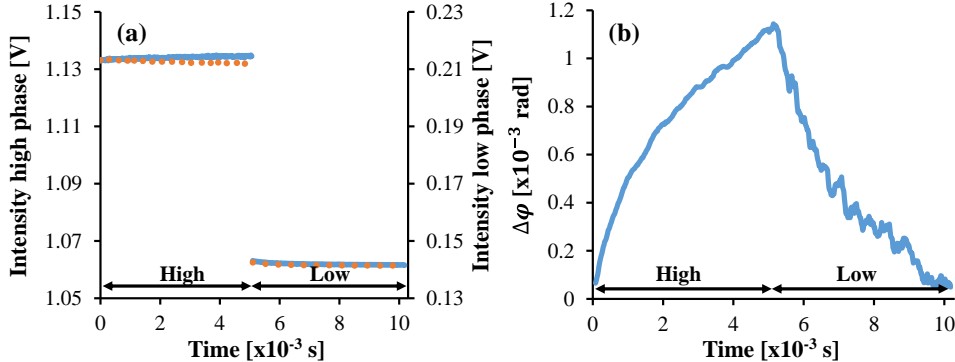

**Figure 4 – (a) Typical raw signals from the interferometer outputs ($I_1, I_2$; blue and red dotted lines in (a), respectively) measured during a modulation cycle with a strongly absorbing sample. Over the course of the heating (high) phase, an intensity difference arises between the outputs. (b) This effect is seen more clearly in the resultant phase shift, which is calculated by normalising the difference of the raw signals by the total intensity (Eq. 5). The baseline offset has been subtracted.**




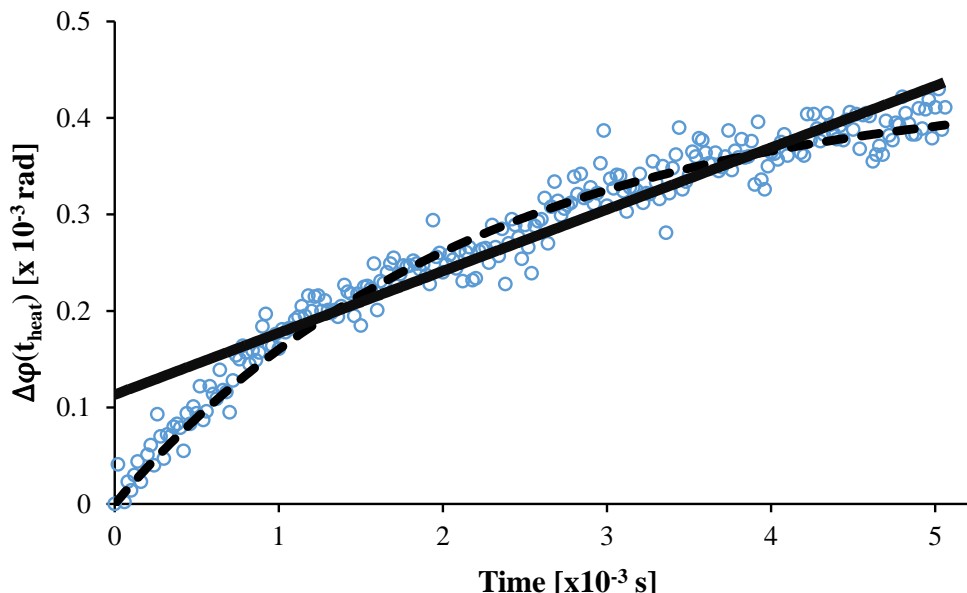

**Figure 5 – Measured phase shift during the heating phase averaged over a 1 second interval for a strongly absorbing aerosol (approximately 100 μg m⁻³ eBC) in argon. The black dashed line represents the best least squares exponential fit, while the dotted line is the linear fit to the data.**

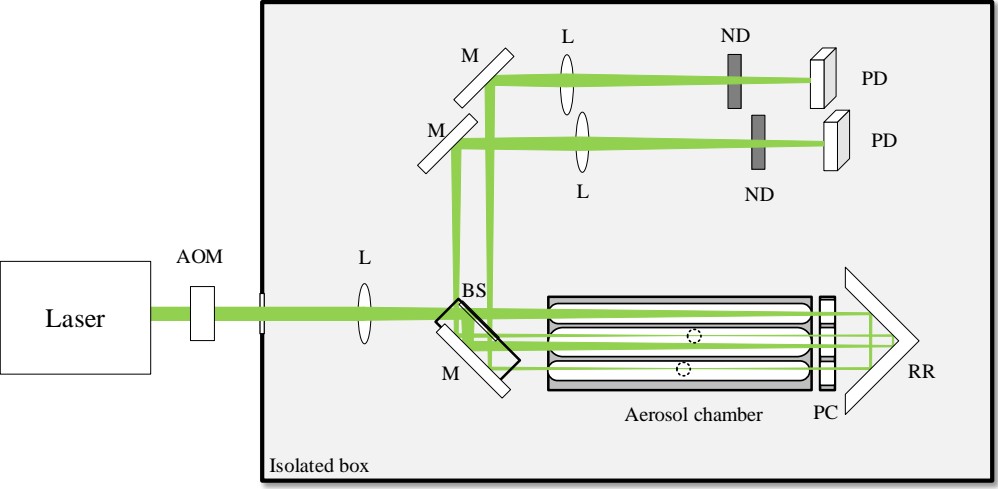

**Figure 6 – Schematic of the MSPTI prototype. The marking M denotes mirrors, while L, AOM, BS, PC, RR, ND and PD denote focusing lenses, acousto-optic modulator, beam splitter, pressure chamber, retroreflector, neutral density filter and photodiode, respectively. The dashed circles show the positions of the beam focus in each arm of the interferometer. An image of the experimental set up is shown in Figure S6 in the Supplementary Information.**



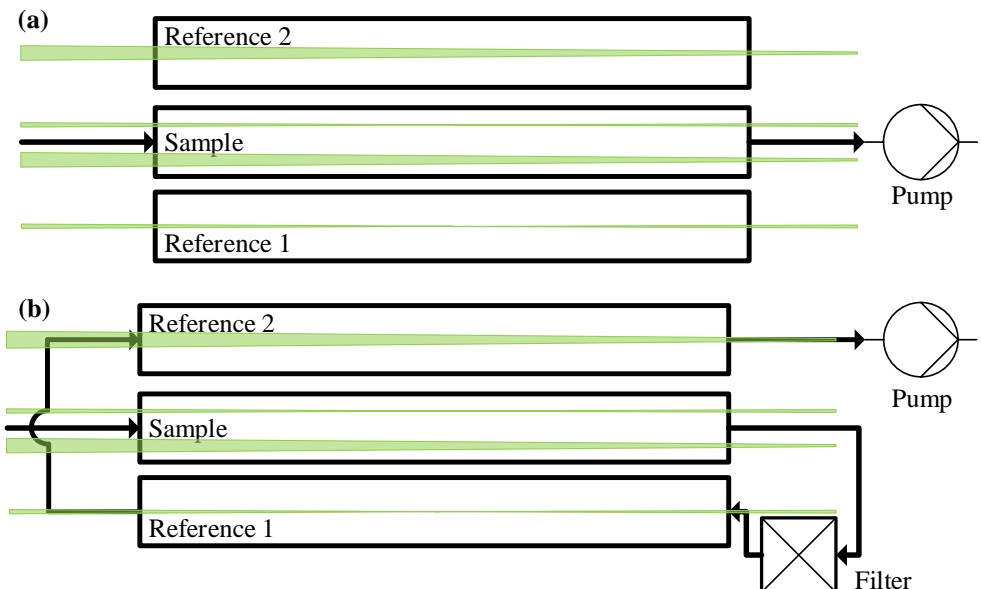


**Figure 7 – Gas flow system for the MSPTI prototype. Calibration measurements are performed as in (a) by filling the sample cell with the calibration gas and the reference cells with non-absorbing synthetic air. All three cells are connected for standard measurements as in (b), with the filtered sample flowing through both reference chambers in sequence.**

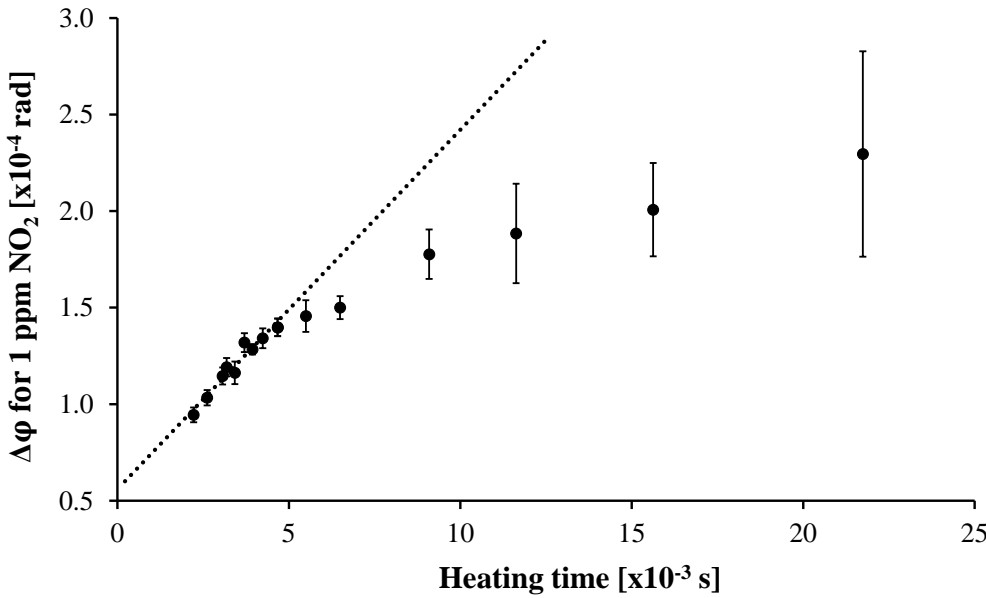


**Figure 8 – MSPTI signal dependence on the heating time for 1 ppm of $NO_2$ in synthetic air. For shorter heating times the PTI signal is linearly dependent on the heating time (dashed line). Error bars represent one standard deviation of the 10 s integrated data.**

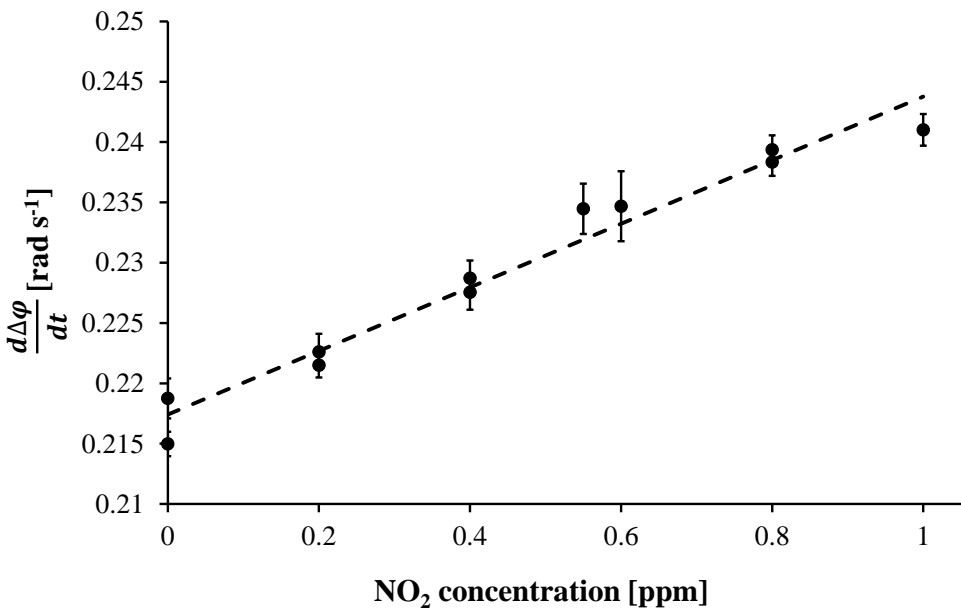


**Figure 9 – PTI signal measured for two consecutive measurement series with NO₂ concentrations between 0.2 and 1 ppm measured at a flow rate of 0.5 lpm. Error bars indicate one standard deviation of the data integrated over 10 seconds. The dashed line represents the best linear fit to the data set.**

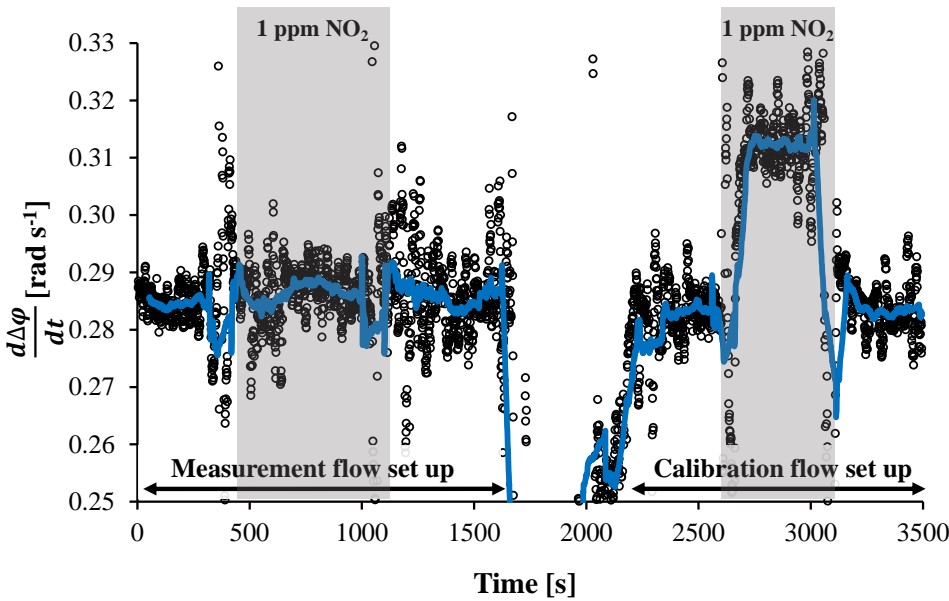


**Figure 10 - Demonstration of the relative nature of the MSPTI measurement. Circles represent averages over 10 seconds, whereas the blue line is the 100 s moving average. In the measurement flow configuration NO₂ is present in both sample and reference chambers and no longer contributes to the MSPTI signal.**

**Table 1: Limits of detection for the MSPTI for different integration times. Measured phase shifts were converted into absorption coefficients using the conversion factor from the NO₂ calibration presented in Figure 9. The eBC**





concentration was calculated with a MAC value of 10 m$^2$ g$^{-1}$. This is lower than the value of 12 m$^2$ g$^{-1}$ obtained by transferring the measurements of ambient BC particles by Zanatta et al. (2016) to 532 nm using an Angström exponent of 1. Electrical discharge generated BC has been shown to have a MAC value lower than that of ambient soot (Schnaiter et al., 2003). As the MAC of the BC from the employed BC generator has not been measured directly, the limits of detection for eBC should be taken as a reference only.

| Averaging time [s] | Standard deviation of MSPTI signal [$10^{-3}$ rad s$^{-1}$] | Limit of detection | | |
|---|---|---|---|---|
| | | b$_{abs}$ [Mm$^{-1}$] | NO$_2$ [ppb] | eBC [ng m$^{-3}$] |
| **1** | 4.13 | 49.6 | 165 | 4960 |
| **10** | 1.22 | 14.6 | 49 | 1460 |
| **60** | 0.61 | 7.35 | 25 | 735 |
| **300** | 0.44 | 5.3 | 18 | 530 |

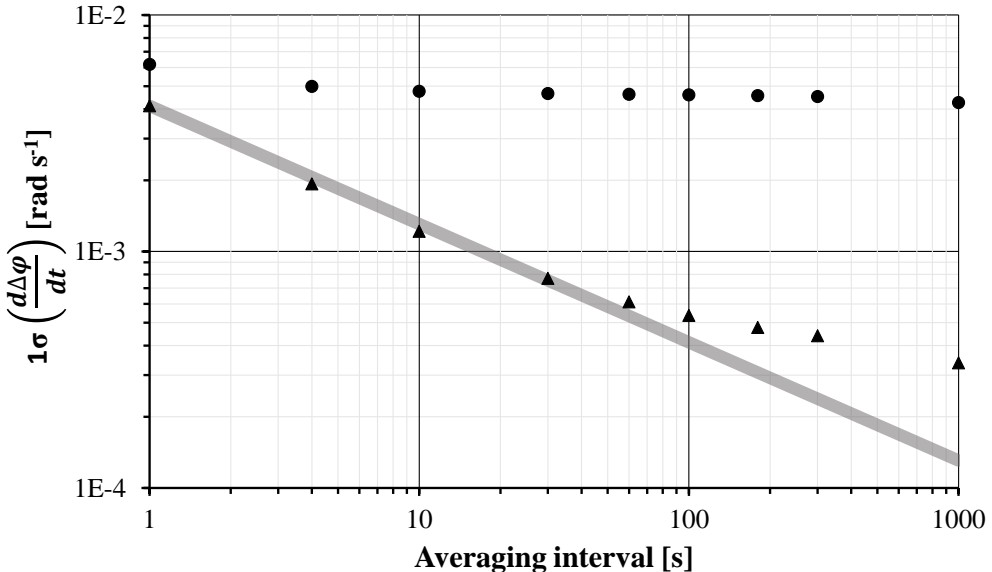

**Figure 11 – The standard deviation of the baseline for drift uncorrected (filled circles) and drift corrected data (filled triangles). The drift corrected data approximately follows a square root dependence (grey line) on the averaging interval up to an averaging interval of 60 seconds. Note the logarithmic scales.**

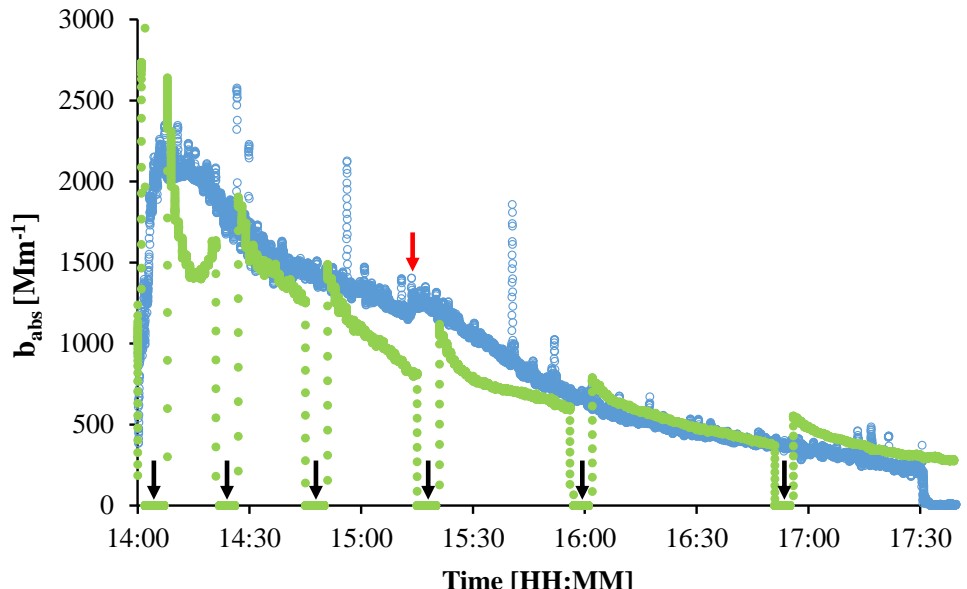

**Figure 12 – Comparison of measured light absorption for the MSPTI (blue) and AE33 (green) sampling from a common**
**pre-prepared aerosol reservoir filled with graphitic soot prepared with an electrical discharge source. Data points**
**represent a 9 second running average of 1 second data. Black arrows indicate automated filter changes for the AE33**
**and the red arrow indicates a mode hop by the MSPTI laser.**