# Peer review of "A single-beam photothermal interferometer for in-situ measurements of aerosol light absorption"

_Atmospheric Measurement Techniques, 2020_

## Referee Comment (RC1) · Anonymous Referee #1 · 31 Jul 2020

In situ aerosol absorption measurements are necessary to understand the climatic and health impacts of BC and BrC aerosols. For decades, aerosol absorption has been measured by filter-based methods that are inexpensive, but suffer from artifacts due modification of the particles collected on the filter and optical interactions between the absorbing particles, non-absorbing particles, and the filter material. Photoacoustic and photo-thermal instruments that measure aerosol absorption while the particles are suspended in the sample air and avoid these artifacts. However, the photoacoustic and photo-thermal methods currently are complex and expensive preventing their broader use. Visser et al. demonstrate an innovative prototype photo-thermal instrument that is simpler than previous designs and could contribute to the broader deployment of

more accurate absorption measurements. The novel optical arrangement of this PTI instrument make is appropriate for publication in AMT, although it is at a mid-point in its development, and further work to improve it detection limit, eliminate artefacts due to laser mode hopping, improve acoustic isolation, and optimize operational parameters (i.e. modulation frequency) will determine its ultimate usefulness.

General Comments: 1) The phase shift as a function of heating time (Figure 5) is nonlinear as the author correct point out, but then they proceed to analyze with a linear fit. The authors should use a nonlinear fit routine and eq. 8 to fit the data. The justification of the linear fit in the supplemental merely demonstrate at the linear fit slope is proportion to the absorption coefficient for a narrow set of conditions. The proportionality could change with bath gas, pressure and possibly RH.

2) This instrument could be understood in the framework of a thermal nonlinear optical effect (Boyd, Nonlinear Optics 3rd Edition 2008, section 4.5). The interferometer mea- sures the self-phase modulation of due to the heating of the aerosol. There will also be a thermal lens. Is it a significant? How would a thermal lens (which could change the path of active arm of the interferometer) affect the phase measurement?

3) Plots of the sensitivity (Figs 9 and 10) are presented in units of radian-seconds. Scaling these plots so the y-axis is in units of absorption (cm-1, Mm-1) would be a more natural unit and help the reader easily compare with other absorption measurements.

4) At times in the manuscript, the instrument is are presented as a measurement of BC concentration, but given the variability and uncertainty of the BC MAC in the ambient atmosphere and the contributions of BrC, it would be better to frame instrument as a quantitative measurement of the absorption coefficient rather than a semi-quantitative measurement of BC concentration.

5) It is not clear why the authors chose to modulate at low frequencies where the heating curve is nonlinear rather than modulate a higher frequencies and avoid the nonlinearity. Are there limitations due to the laser or AOM?

Specific Comments:

Line 49: The MAC =10 is reasonable, but it is not extrapolated with AAE = 1.

Lines 155-185: rather long explanation, could be tightened up a bit.

Figure 5: the dotted line looks like a solid line to me.

Line 293: what is the sensitivity to non-50:50 BS. Typically, precision on commercial BS is not great and can vary with polarization and angle of incidence.

Line 421: Does the filter give a pressure drop between the sample and reference cells? Is this accounted for in the PC volumes?

Figure 9: maybe color the points differently for the ramp up and down in NO2 concentration, so the outset is clear.

Line 439: The need to monitor the baseline drift negates the advantaged pointed out in the previous paragraph (Lines 416 -431).

Figure 11: Maybe this should be replaced with an Allan deviation plot which is appropriate to differentiate between short-term precession and long-term drift

Line 492-495: Several photoacoustic absorption measurements use active charcoal scrubbers very effectively to remove gas-phase absorbers before measurement of the aerosol absorption.

---

## Referee Comment (RC2) · Anonymous Referee #2 · 4 Aug 2020

This manuscript introduces an instrument that utilizes a modulated single-beam photothermal interferometer (MSPTI) to measure the aerosol light absorption. The topic of this manuscript is completely relevant to the aim and scope of AMT. The reviewer enjoys reading about the analysis for the phase shift, and strongly recommends the publication. Then, the reviewer thinks that this manuscript is ready for publication and can be accepted for publication in AMT only after some minor revisions are performed. The detailed points for revision in this paper are shown as below;

1. Introduction seems to be a little bit longer than what it ought to be. Is it important in this study to mention the vertical measurement of black carbon? If possible, please

[Figure]

consider shrinking it into around 2 pages.

2. As a matter of fact, one of the important things in the ambient measurement is the durability. Is the future experimental setup able to monitor the aerosol light absorption for longer than 24 hours?

3. The reviewer is wondering if the authors checked the intensity of the laser power would be 50:50 after the beam-splitter. It would be great to leave a comment on the performance of the beam-splitter such as a plot of transmittance vs. wavelength. (Please see the graphs at https://www.thorlabs.com/newgrouppage9.cfm?objectgroup_id=914)

4. Is it possible for the authors to say the power of laser in the current setup? The reviewer can only imagine it from the last paragraph of RESULT section (It must be less than 400 mW).

5. Line 63: One of the hallmarks of this study is to measure the optical property of light absorbing aerosols in an airborne state. In this sense, the reviewer recommends that the authors append a reference (Lee, 2019) to provide the drawback of any filter-based techniques.

6. Line 238: Please make it sure whether the number of equation is correct or not. Equation 3? Or equation 5?

7. Line 252: Please replace 'below a characteristic value' with 'shorter than a characteristic time'

8. Line 286: Is the absolute filter HEPA-grade?

9. Line 351-356: "PTI is an in situ light absorption $\sim$ and the resultant PTI signal.". This is a general explanation about calibration, thus it is irrelevant to appear in Result section. The reviewer recommends that the sentences be moved to Experimental section, maybe at Line 349.

10. Line 376-380: This paragraph is unclear. Please reword the sentences so that

potential readers may understand what it is.

11. Line 456: The MAC and the filter multiple-scattering enhancement parameter were provided from the user's manual of AE33. Is it possible for the authors to comment on how 13.14 m2/g and 1.57 were derived for the MAC and the multiple scattering parameter, respectively?

References Lee J. Performance Test of MicroAeth® AE51 at Concentrations Lower than 2 $\mu$g/m3 in Indoor Laboratory. Applied Sciences. 2019, 9(13), 2766.

---

## Referee Comment (RC3) · Anonymous Referee #3 · 10 Aug 2020

Manuscript: A single-beam photothermal interferometer for in-situ measurement of aerosol light absorption (Visser et al.,)

Quantifying aerosol light absorption with high precision and accuracy remains an elusive but important need the climate change community trying to quantify the contribution of aerosols to the Earth's radiation budget. Measurement of light absorption is typically conducted using filter-based techniques which offer very high precision but are greatly hindered in accuracy due, in part, to their well-known measurement bias' which cannot be completely removed. Thus there exists a need to develop in situ techniques that can directly measure light absorption. Photothermal interferome-

try (PTI), like it photoacoustic cousin, is one class of measurement methodology can addresses this need by directly measuring aerosol light absorption through the dissipation of spectrally-absorbed energy by a particle. Another important hallmark of these photo thermal approaches is that their complete insensitivity to light scattering, which is a considerable advantage given that optical extinction by particles is dominated by scattering, save, of course, pure soot (e.g., black carbon).

The manuscript by Visser et al., describes a novel single-beam photothermal interferometer - called the modulated single beam PTI (or MSPTI) - to directly measure aerosol light absorption. The unique and clever wrinkle of the MSPTI is the single beam design that serves as both the interferometer (probe) laser AND as the excitation (pump) laser. Compared to its two-beam brethren - which uses a dedicated interferometer laser (typically a single-frequency HeNe laser) and a separate pump laser (405 nm, 532 nm, or 670 nm being typical wavelengths of interest) the single-beam configuration offers the significant improvement in the ease of instrument laser alignment. An additional advantage of the single-beam interferometer is the ability to remove, in realtime, signal contributions by light absorbing molecules by having the reference arm sample filter-free air. In contrast, two-beam PTIs require periodic acquisition of particle-free conditions via HEPA-filtered air to obtain a measure of the molecular contribution. Another unique advance in the Visser et al. design is the use of a pressure cell to help maintain phase quadrature lock as this feedback mechanism eliminates moving parts within the instrument thereby improving instrument robustness.

While the performance metrics for this "proof-of-principle" version of the MSPTI currently relegate its immediate utility to long-term, fixed ground-based measurements (for a 60-second integration a lower absorption detection limit $\sim$7.5 Mmˆ-1 or, assuming a mass absorption cross-section of 10 mˆ2/g, an equivalent black carbon (eBC) concentration of 0.75 ug/mˆ3), the instrument is a worthy addition to the measurement quiver in the aerosol direct effects community. With a couple of exceptions noted below, this manuscript is clearly written and requires only minor revisions. It is therefore
recommended that this manuscript be published.

Page 6, line 11. What was the criteria for selecting I_low and I_high? All that is stated is that the laser "is modulated between two sufficiently different intensity levels." Certainly there is a lower limit below which the ability to lock on to quadrature would be compromised. On the other hand, the larger the difference, would favor signal detection.

Page 8, lines 8 and 9: the use of "solid" and "external" noise sources is not very descriptive. Why not call this noise sources what they actually are: mechanical (vibrational) noise and acoustic noise. For those unfamiliar with PTI or, more generally interferometry, referring to a noise source as "solid" or "external" is a bit nebulous.

Page 8, Lines 15-17. The MSPTI utilizes a reference channel that samples filtered air - a necessary condition for the single-beam configuration to work. What is the impact of a sample containing a mixture of light absorbing and non-light absorbing particles at high concentrations, as might be encountered in a biomass burning event, where the refractive index (RI) of the particles could contribute to the sample ensemble RI but whose contributions would not be present in the particle filtered sample? What are thermal lensing implications under these conditions? In a two-beam PTI, the sample and reference arms probe the same particle-laden air simultaneously thereby enabling common mode rejection for such conditions.

Page 10, line 4. The authors are encouraged to cite Lack et al. (2006) here. This paper is already listed in their citations.

Page 11, line 16. The authors are encouraged to merge Figures 5 and 8. In a lot of ways, Figure 8 is far more informative as it beautifully captures how decreasing the modulation frequency - increasing the heating period - brings about significant departure from linearity due to energy diffusion outside the probe region.

Page 11, Line 22. The sentence "If the deviation from linearity of the PTI signal with

heating time due to heat loss out of the measurement volume could be excluded from the measurement, then both the measured signal and, by extension, the sensitivity determined from calibration measurements would be considerably higher." This is a very awkwardly worded sentence. I believe that the authors simply trying to say that the non-linear signal due to diffusional loss of heat outside the probe volume suppresses measurement of the total amount of energy deposited into the system. If so, please clarify. [As an aside from purely physics interest, this raises an interesting question with respect to a two-beam PTI: if the probe volume was configured to be slightly larger than the pump volume would this enable the 2-beam configuration to "delay" the onset of the departure from linearly and, in so doing, improve performance at lower modulation frequencies? ]

Page 11, line 36. It seems to this reviewer that the two time series traces should be switched. Ideally, the authors should first show that their system can indeed detect NO2 (currently the right most trace) and THEN show how well their system does at removing the NO2 signal (currently the left most trace). The actual time stamps is immaterial here. This is a stylistic comment.

Page 12, line 2: The authors are reminded that there are chemical "denuders" for removing molecular species such as NO2 via MnO2.

Page 12, Comparison with Aethalometer. Philosophically, this reviewer has major concerns about the underlying assumption of a constant mass absorption cross-section (MAC) for black carbon (BC) in order to report an equivalent black carbon (eBC) concentration. There are a plethora of studies showing that the BC MAC (at 550 nm, for example) can vary from ∼7.5 m2/g for uncoated BC particles to 13-15 m2/g for coated particles. This reviewer understands that the Aethalometer reports a eBC value and that the authors are comparing their instrument to the Aethalometer. While this comment is well-beyond the scope of this present paper, one potential (and easy) solution that the authors might consider, is to compare absorption coefficients instead of mass concentrations - after all, this is what both instruments fundamentally measure.

Page 23, Figure 11. Is the departure observed in the variation from t^-1/2 due to the active quadrature lock feedback circuit? Also a more meaningful metric to the aerosol community would be an Allan variance plot of the absorption coefficient.

---

## Author Comment (AC1) · 26 Sep 2020

The authors thank the reviewers for their time in reviewing the manuscript and their constructive questions and comments. The manuscript will most certainly be improved by implementing the suggested changes. On a personal level, the authors also very much enjoyed the high level discussion of our instrument and PTI in general.

Addressing the comments and questions of Anonymous Reviewer #1 in order:

1) Previous iterations of PTI instruments have relied upon lock-in detection for the evaluation of the PTI signal. With lock-in detection the signal is multiplied by a reference

sine wave and the amplitude component (R) is interpreted as the PTI signal. Information regarding the shape of the heating curves could potentially be extracted from the phase signal provided by the lock-in amplifier, though the authors are not aware of any PTI study of aerosols in which the phase of the PTI signal has been examined. It is assumed that the shape of the heating curve does not change with absorption. If this assumption is correct then the instrument response is linearly dependent on absorption (Sedlacek 2006, Figure 3). In the current work, the linear fit to the heating curves was employed as a simple alternative to lock-in detection. The linear fit is equivalent to lock-in detection. The measurement of the lock-in amplifier is the product (in the space of orthogonal sine functions) of the heating curve and the sine with the same frequency. If the shape of the heating curve changes, this will be observed in the amplitude of the product (and thus potentially interpreted as a change of absorption) and the measured signal phase. Therefore, unless the phase of the PTI signal with respect to the laser excitation is investigated, linearity (or at least a consistent heating curve form) is assumed and both lock-in detection and the linear fitting of the PTI data are equivalent. The authors acknowledge that a change in the shape of the heating curve would lead to a change in the proportionality of the linear and non-linear fits. It is also agreed, that this proportionality could change with a significant change in the composition of the bath gas, pressure and potentially RH, which would all result in considerably different thermal properties of the sample air stream. These effects have so far not been observed in the experiments. We aim to investigate the proportionality in a future work and examine whether additional information can be extracted from the sample.

2) This is correct. It is expected that a thermal lens forms due to the temperature distribution in the bath gas along the laser path, in particular around the laser focus and affects the paths of the laser beams through the instrument. The effect of a thermal lens would be two fold; firstly it would change the optical path length of the interferometer arm, thus influencing the phase measurement, Secondly, it would alter the size of the laser beam focus, which feeds back into the PTI measurement and the thermal lens. Experimentally, the authors have not been able to observe the effects of a thermal

lens or at least decouple this effect from, for example, the non-linearity observed in the heating curves due to loss of heat out of the measurement volume during the heating phase. In comparison to previous instruments and their response, it is believed that the energy density in the laser focus in the current study is very similar and that the effect of a thermal lens would be comparable in each case.

3) In choosing the units for presentation of the data, the authors referred to previously published works on photothermal interferometry, such as (Sedlacek 2006). Here equivalent plots are presented in terms of (fractions of) volts, which is the direct measurement unit in the case where a lock-in amplifier is used. Figure 9 presents the response of the MSPTI signal due to the addition of 1 ppm NO2 in both flow schemes. As the instrument is calibrated with NO2, the authors feel that it would be best to present this plot with two y-axes, one with the internal PTI units (rad s-1) and the other with units of absorption (Mm-1). The plot will be updated in the manuscript to add this second y-axis. In Figure 10, the standard deviation of a background measurement is presented. The y-axis of this plot will be changed in the manuscript to be in terms of absorption (Mm-1).

4) The reviewer makes a very good point. The instrument does indeed measure light absorption. References to black carbon were only meant for comparison purposes and to facilitate an easier understanding of the quantities of BC measured by presenting the measured values as a mass concentration by assuming a stable MAC. The manuscript will be changed to be more clearly framed in terms of the measurement of absorption, with eBC as a secondary parameter.

5) We have chosen the frequency based on the frequency dependence of the signal-to-noise ratio. The maximum of this function lies at 91 Hz. If the instrument were reliant on intensity modulation by changing the current applied to the laser head, then yes, even reaching 91 Hz would not have been possible. The AOM however, allows much higher modulation frequencies, even approaching 1 MHz. The non-linearity seen in the plot of PTI signal vs heating period (Figure 8) begins for heating periods above approximately

5 ms. We attempted experiments with the heating times slightly below this (modulation frequencies between 100 and 130 Hz) but encountered significantly increased noise in this frequency range due to other lab equipment. 91 Hz was subsequently chosen as it provided the optimum signal to noise ratio for the current instrument and the calibration was not required to be transferred to a different modulation frequency.

Addressing the specific comments of Anonymous Reviewer #1:

- With regards to the line 49 and the MAC not being extrapolated with AAE=1 The reviewer is correct. This line of text will be corrected in the manuscript.

- With regards to the text on lines 155-185 being a rather long explanation The authors agree with the reviewer. The text from lines 155-185 will be reworked in the manuscript to improve the readability.

- With regards to Figure 5 and dotted vs solid line Thank you to the reviewer for finding this error. The caption in the manuscript will be updated from dotted to solid line.

- With regards to line 293 and the precision of commercial beam splitters The instrument shows considerable sensitivity to non-50:50 beam splitting. A large variance has been observed for the polarisation angle. The sensitivity to angle of incidence has not been investigated as the very little variation is possible given the design of the instrument. The polarisation of the laser beam was adjusted to ensure 50:50 splitting of the beam as measured using a power meter as well as the interferometric contrast. It was found that the highest interferometric contrast was obtained when the laser beam intensity was split very close to 50:50, due to equal losses in both beam paths. Subsequent use of a 532 nm laser-line beam splitter that is much less polarisation sensitive has shown that the drift of the baseline measurement is not due to changes in the splitting ratio of the beam splitter (i.e. the splitting ratio appears to remain stable during the measurements).

- With regards to line 421 and the pressure drop across the filter Yes, the use of a

filter did result in a pressure drop of around 1 mBar from the measurement chamber to the reference chamber. This was not accounted for in the volumes of the pressure chambers. In recent work we have instead changed the gas line configuration such that there is no longer a pressure difference between the sample and reference chambers.

- With regards to Figure 9 and colouring the data points differently This is a good suggestion to improve the clarity of Figure 9. The Figure will be updated in the manuscript to have different coloured points for the ramp up and down measurements.

- With regards to line 439 and the necessity of measuring the background drift The authors respectfully disagree with this statement. There are two potential sources of the background change: gas absorption and photothermal effects in the optical elements in the interferometer. Two-beam interferometers must in theory measure both changes or resort to another way of determining the concentration of gaseous species or employ e.g. a scrubber. We account for this with the reference chamber. The absolute baseline must continuously be monitored in any PTI instrument as it arises primarily from light absorption by optical elements in the interferometer, which may change over time. This type of baseline change is different to that arising from the absorption due to gaseous species and changes thereof. Two beam configurations with glancing angle type configurations are rather sturdier in this respect. Our single beam configuration features a perfect overlap between the pump and the probe beams – they are the same laser beam, but this is also true for the effects in the optical elements. Hence the need to measure the background.

- With regards to Figure 11 and the Allan deviation The authors feel that the standard deviation is more appropriate than the Allan deviation in expressing the uncertainty in PTI measurements. It is however acknowledged that presentation of the Allan deviation would better enable comparison to other measurement techniques. A plot of the Allan deviation using the same data as Figure 11 will be added to the supplementary information.

- With regards to lines 492-495 and charcoal scrubbers This is correct and a note to this effect will be added to the manuscript. It is however always advantageous to treat the aerosol as little as possible before measurement, in order to avoid changing any of its characteristics.

References Sedlacek, A., Real-time detection of ambient aerosols using photothermal interferometry: Folded Jamin interferometer, Rev. Sci. Inst., 2006, 77, 064903
* * *

---

## Author Comment (AC2) · 26 Sep 2020

The authors thank the reviewers for their time in reviewing the manuscript and their constructive questions and comments. The manuscript will most certainly be improved by implementing the suggested changes. On a personal level, the authors also very much enjoyed the high level discussion of our instrument and PTI in general.

Addressing the specific comments and questions of Anonymous Reviewer #2:

1) The authors acknowledge that the introduction is a little lengthy. It will be reworked slightly for brevity.

[Figure]

2) This is indeed a very important requirement for use of the instrument in ambient measurements. Recent tests have shown that the instrument can be operated for more than 24 hours without issue and the future instrument should be durable enough to measure for months on end to fulfil its function as an ambient monitoring instrument. A sentence will be added to the introduction of the manuscript to stress this point.

3) Yes, the authors have investigated the splitting ratio of the beam splitter for various polarisations of the incoming laser beam and have seen a significant dependence of the splitting ratio on the polarisation. This was measured both using a power meter and by determining the contrast of the interferometer. We cannot comment on the wavelength dependence of the beam splitter as all of the measurements were performed at a single wavelength. Subsequent testing has shown that the splitting ratio and polarisation insensitivity is much improved for the 532 nm laser line beam splitter as compared with the broad-band version.

4) The maximum power of the laser is 450 mW. The power employed in the study was 200 mW as at the time the cooling was insufficient to run the laser at higher powers for extended periods. The laser power employed in the study will be added to the experimental section of the manuscript.

5) Thank you for the reference. It will be added to the introduction of the manuscript.

6) Equation 3 is correct. It shows how the differential phase is calculated from the difference of intensities at the detectors divided by the total light intensity.

7) Thank you. The manuscript will be changed for this improved phrasing.

8) Yes, the absolute filter is HEPA grade. The manuscript will be updated to reflect this.

9) The authors accept the recommendation and the lines in question will be moved to the end of the experimental section.

10) The authors intended to bring across the message that the heating curves were not observed to be linear, including for measurements made with heating periods in the

so-called linear range seen in Figure 8. The shapes of the heating curves remained constantly non-linear in this range, however, thus leading to the linear relationship between the PTI signal and the heating time in this range. This section of text will be updated for clarity. Please see also replies to Anonymous Reviewer 1.

11) Filter photometers are calibrated using the mass attenuation cross-section (Gundel et al., 1984). The mass attenuation cross-section is a product of the mass absorption cross-section and the filter multiple-scattering parameter, using the parameterization of Weingartner et al. (2003). Drinovec et al. (2015) have determined the C for the AE33 filter (at that time) relative to the value from Weingartner et al. (2003). Filter photometer response to a complex sample with a high SSA is more complicated (Lee, 2019) – the scattering of the sample affects the measurements (Weingartner et al., 2003; Arnott et al., 2005) and this cross-sensitivity to scattering affects the measurement. This is often measured as a change in the effective (apparent) multiple-scattering parameter, that is the slope between the reference absorption measurement and the filter photometer. We do not observe this effect as the SSA of our aerosol samples is very low.

References: Gundel, L. et al., The relationship between optical attenuation and black carbon concentration for ambient and source particles, Sci. Total Environ., 1984, 36, 197 Weingartner, E. et al., Absorption of light by soot particles: determination of the absorption coefficient by means of aethalometers, J. Aerosol Sci., 2003, 34(10), 1445 Arnott, W. P., et al., Towards Aerosol Light-Absorption Measurements with a 7-Wavelength Aethalometer: Evaluation with a Photoacoustic Instrument and 3-Wavelength Nephelometer, Aerosol Sci. Tech., 2005, 39(1), 17 Drinovec, L. et al., The "dual-spot" Aethalometer: an improved measurement of aerosol black carbon with real-time loading compensation, Atmos. Meas. Tech., 2015, 8(5), 1965

---

## Author Comment (AC3) · 26 Sep 2020

The authors thank the reviewers for their time in reviewing the manuscript and their constructive questions and comments. The manuscript will most certainly be improved by implementing the suggested changes. On a personal level, the authors also very much enjoyed the high level discussion of our instrument and PTI in general.

Addressing the comments and questions of Anonymous Reviewer #3 in order:

- With regards to page 6, line 11 and the criteria for selecting I_low and I_high: Yes, the reviewer is completely correct and a larger difference between I_low and I_high leads

to an increase in signal. The initial selection of I_low was determined by the accuracy at which the signal during the cooling (or low) phase could be determined. As this data was used to lock quadrature and provide some qualitative indication of the signal during the cooling phase, a minimum laser power of 20 mW was required to provide acceptable signal to noise. It should however be noted, that it is possible to lock quadrature with only the heating phase (or high) signals, thus making it possible for I_low to be zero, if no information about the cooling phase is required. However, with the AOM we have so far only been able to achieve a 20x reduction in the laser intensity in the main beam and therefore we have not been able to test this without reconfiguring the outputs of the AOM. Additional information will be added to the manuscript to explain this.

- With regards to page 8, lines 8 & 9 and the description of the noise sources: The authors agree on this point and the manuscript will be updated to make it clearer in this respect.

- With regards to page 8, lines 15-17 and the effect of particle RI contributions at high concentrations: This is a very interesting question and one that will require some further study to experimentally validate. In theory, PTI measurements are only sensitive to changes in the measured refractive index at the modulation frequency. This means that static differences in refractive index, as well as slow changes in the refractive index (even relative changes) should not affect the measured signal, regardless of the source. Even at very high concentrations, where the RIs of the various particles contribute in a meaningful way to the ensemble, the authors do not believe that any significant artefacts due to a static difference in RI will be present. At such high concentrations however, light attenuation by the sample will be large and therefore the interferometric contrast will suffer, reducing the accuracy of the measurements. We have not tested the MSPTI at sufficiently high concentrations to investigate this, but there is almost certainly an upper concentration limit beyond which the measurements are no longer valid. The implications for thermal lensing are the same as for the twobeam PTI configurations. A thermal lens will be formed in the sample beam, which is the equivalent of the sample beam / pump beam thermal lens in the two-beam PTI configuration (with some differences due to the beam geometries). In the absence of light absorbing gases, no thermal lens will be formed in the reference beam, just as for a two-beam configuration. It is however slightly different in the case where an absorbing gas is present in the reference chamber. Under these conditions a thermal lens will be formed in the reference beam of the MSPTI, whereas none would be formed in the two-beam PTI. However, since the absorption of the gas is the same in both sample and reference chambers, the contribution to the thermal lens from the gas absorption in both chambers should also be equal.

- With regards to page 10, line 4 and the citation: The citation will be added to this line of the manuscript.

- With regards to page 11, line 16 and the comparison of Figures 5 and 8: This is unfortunately confusing, however Figures 5 and 8 don't show the same data or effect. Even the heating curves for the data points in the linear dependence region of Figure 8 are not themselves linear. The linear dependence of the PTI signal on the heating period in Figure 8 only shows that the shape of the heating curves has remained consistent within this range. The discrepancy between the heating curve and the linear fit outside of this range increases with increasing heating period, causing the nonlinearity of the PTI signal as a function of the heating period (Figure 8).

- With regards to page 11, line 22 and the clarification of text: Yes, thank you for helping to clarify this section of text. The manuscript will be updated to clarify this point. The saturation should occur at longer times by increasing the diameter of the probe beam, assuming no other cause of non-linearity becomes dominant. However, the signal strength is the intensity weighted average of the phase shift across the beam cross-section. The signal at shorter heating times will be reduced due to the time it takes for the heat from the absorption process to be conducted into the entire probe beam volume.

- With regards to page 11, line 36 and the reordering of the data presented in the figure: Yes, the authors concur that this would be preferential, both in a scientific as well as aesthetic manner. The figure will be updated in the manuscript.

- With regards to page 12, line 2 and the availability of denuders: This is correct. The authors were trying to make the point that it is possible for the MSPTI to measure aerosol particle absorption in the presence of absorbing gases without the need to modify the sample. The manuscript will be updated to clarify this point and the availability of denuders.

- With regards to page 12 and the comparison with the Aethalometer: This is correct. The mass concentrations were only intended for comparison with other measurements and to give aerosol scientists a metric to better understand the measured absorption (i.e. mass concentrations). The manuscript will be updated to focus more on absorption and use eBC as a secondary metric to clarify this point.

- With regards to page 23, Figure 11, sources of noise and the Allan deviation: As the quadrature lock circuit was operating at a frequency of 1 Hz or below, one source of the deviation from the inverse square ideal line could be partially due to the constant adjustments from this circuit. The main contribution however seems to be the low frequency drifts in the baseline. The authors feel that the standard deviation is more appropriate than the Allan deviation in expressing the uncertainty in PTI measurements. It is however acknowledged that presentation of the Allan deviation would better enable comparison to other measurement techniques. A plot of the Allan deviation using the same data as Figure 11 will be added to the supplementary information.

––––––––––––––––––––––––––––––––––––––

---

## Author Response (AR1)

**Authors' response**

The authors thank their reviewers for their time in reviewing the manuscript and their constructive questions and comments. The manuscript will most certainly be improved by implementing the suggested changes. On a personal level, the authors also very much enjoyed the high level discussion of our instrument and PTI in general.

**Reviewer 1**

1) **The phase shift as a function of heating time (Figure 5) is nonlinear as the author correct point out, but then they proceed to analyze with a linear fit. The authors should use a nonlinear fit routine and eq. 8 to fit the data. The justification of the linear fit in the supplemental merely demonstrate at the linear fit slope is proportion to the absorption coefficient for a narrow set of conditions. The proportionality could change with bath gas, pressure and possibly RH.**

Previous iterations of PTI instruments have relied upon lock-in detection for the evaluation of the PTI signal. With lock-in detection the signal is multiplied by a reference sine wave and the amplitude component (R) is interpreted as the PTI signal. Information regarding the shape of the heating curves could potentially be extracted from the phase signal provided by the lock-in amplifier, though the authors are not aware of any PTI study of aerosols in which the phase of the PTI signal has been examined. It is assumed that the shape of the heating curve does not change with absorption. If this assumption is correct then the instrument response is linearly dependent on absorption (Sedlacek 2006, Figure 3).

In the current work, the linear fit to the heating curves was employed as a simple alternative to lock-in detection. The linear fit is equivalent to lock-in detection. The measurement of the lock-in amplifier is the product (in the space of orthogonal sine functions) of the heating curve and the sine with the same frequency. If the shape of the heating curve changes, this will be observed in the amplitude of the product (and thus potentially interpreted as a change of absorption) and the measured signal phase. Therefore, unless the phase of the PTI signal with respect to the laser excitation is investigated, linearity (or at least a consistent heating curve form) is assumed and both lock-in detection and the linear fitting of the PTI data are equivalent.

The authors acknowledge that a change in the shape of the heating curve would lead to a change in the proportionality of the linear and non-linear fits. It is also agreed, that this proportionality could change with a significant change in the composition of the bath gas, pressure and potentially RH, which would all result in considerably different thermal properties of the sample air stream. These effects have so far not been observed in the experiments.

We aim to investigate the proportionality in a future work and examine whether additional information can be extracted from the sample.

*No changes have been made to the manuscript.*

2) **This instrument could be understood in the framework of a thermal nonlinear optical effect (Boyd, Nonlinear Optics 3rd Edition 2008, section 4.5). The interferometer measures the self-phase modulation of due to the heating of the aerosol. There will also be a thermal lens. Is it a**

**significant? How would a thermal lens (which could change the path of active arm of the interferometer) affect the phase measurement?**

This is correct. It is expected that a thermal lens forms due to the temperature distribution in the bath gas along the laser path, in particular around the laser focus and affects the paths of the laser beams through the instrument. The effect of a thermal lens would be two fold; firstly it would change the optical path length of the interferometer arm, thus influencing the phase measurement, Secondly, it would alter the size of the laser beam focus, which feeds back into the PTI measurement and the thermal lens.

Experimentally, the authors have not been able to observe the effects of a thermal lens or at least decouple this effect from, for example, the non-linearity observed in the heating curves due to loss of heat out of the measurement volume during the heating phase.

In comparison to previous instruments and their response, it is believed that the energy density in the laser focus in the current study is very similar and that the effect of a thermal lens would be comparable in each case.

*No changes have been made to the manuscript.*

3) **Plots of the sensitivity (Figs 9 and 10) are presented in units of radian-seconds. Scaling these plots so the y-axis is in units of absorption (cm-1, Mm-1) would be a more natural unit and help the reader easily compare with other absorption measurements.**

In choosing the units for presentation of the data, the authors referred to previously published works on photothermal interferometry, such as (Sedlacek 2006). Here equivalent plots are presented in terms of (fractions of) volts, which is the direct measurement unit in the case where a lock-in amplifier is used. Figure 9 presents the response of the MSPTI signal due to the addition of 1 ppm $NO_2$ in both flow schemes. As the instrument is calibrated with $NO_2$, the authors feel that it would be best to present this plot with two y-axes, one with the internal PTI units ($rad\ s^{-1}$) and the other with units of absorption ($Mm^{-1}$). The plot will be updated in the manuscript to add this second y-axis.

In Figure 10, the standard deviation of a background measurement is presented. The y-axis of this plot will be changed in the manuscript to be in terms of absorption ($Mm^{-1}$).

*Figure 9 has been updated with a second axis with units of $Mm^{-1}$. The y-axis of Figure 10 has been updated to be in terms of absorption units ($Mm^{-1}$).*

4) **At times in the manuscript, the instrument is are presented as a measurement of BC concentration, but given the variability and uncertainty of the BC MAC in the ambient atmosphere and the contributions of BrC, it would be better to frame instrument as a quantitative measurement of the absorption coefficient rather than a semi-quantitative measurement of BC concentration.**

The reviewer makes a very good point. The instrument does indeed measure light absorption. References to black carbon were only meant for comparison purposes and to facilitate an easier understanding of the quantities of BC measured by presenting the measured values as a mass concentration by assuming a stable MAC.

The manuscript will be changed to be more clearly framed in terms of the measurement of absorption, with eBC as a secondary parameter.

*The text has been updated to focus on the measurement of absorption, with equivalent mass concentrations of black carbon (eBC) provided as a secondary metric for comparison and understanding purposes. The figures have also been updated to present the data in terms of absorption units (Mm$^{-1}$).*

5) **It is not clear why the authors chose to modulate at low frequencies where the heating curve is nonlinear rather than modulate a higher frequencies and avoid the nonlinearity. Are there limitations due to the laser or AOM?**

We have chosen the frequency based on the frequency dependence of the signal-to-noise ratio. The maximum of this function lies at 91 Hz.

If the instrument were reliant on intensity modulation by changing the current applied to the laser head, then yes, even reaching 91 Hz would not have been possible. The AOM however, allows much higher modulation frequencies, even approaching 1 MHz.

The non-linearity seen in the plot of PTI signal vs heating period (Figure 8) begins for heating periods above approximately 5 ms. We attempted experiments with the heating times slightly below this (modulation frequencies between 100 and 130 Hz) but encountered significantly increased noise in this frequency range due to other lab equipment. 91 Hz was subsequently chosen as it provided the optimum signal to noise ratio for the current instrument and the calibration was not required to be transferred to a different modulation frequency.

*No changes have been made to the manuscript.*

**Specific Comments:**

**Line 49: The MAC =10 is reasonable, but it is not extrapolated with AAE = 1.**

The reviewer is correct. This line of text will be corrected in the manuscript.

*The text has been updated to read: For typical ambient BC aerosols measured at λ= 637 nm the MAC is approximately 10 m$^2$ g$^{-1}$*

**Lines 155-185: rather long explanation, could be tightened up a bit.**

The authors agree with the reviewer. The text from lines 155-185 will be reworked in the manuscript to improve the readability.

*The text between lines 155-185 has been reworked for clarity.*

**Figure 5: the dotted line looks like a solid line to me.**

Thank you to the reviewer for finding this error. The caption in the manuscript will be updated from dotted to solid line.

*This is an issue with the presentation of the image files in the .pdf. The dotted line has been changed to a solid line and the caption has been updated to match.*

**Line 293: what is the sensitivity to non-50:50 BS. Typically, precision on commercial BS is not great and can vary with polarization and angle of incidence.**

The instrument shows considerable sensitivity to non-50:50 beam splitting. A large variance has been observed for the polarisation angle. The sensitivity to angle of incidence has not been investigated as the very little variation is possible given the design of the instrument. The polarisation of the laser beam was adjusted to ensure 50:50 splitting of the beam as measured using a power meter as well as the interferometric contrast. It was found that the highest interferometric contrast was obtained when the laser beam intensity was split very close to 50:50, due to equal losses in both beam paths.

Subsequent use of a 532 nm laser-line beam splitter that is much less polarisation sensitive has shown that the drift of the baseline measurement is not due to changes in the splitting ratio of the beam splitter (i.e. the splitting ratio appears to remain stable during the measurements).

*No changes have been made to the manuscript.*

**Line 421: Does the filter give a pressure drop between the sample and reference cells? Is this accounted for in the PC volumes?**

Yes, the use of a filter did result in a pressure drop of around 1 mBar from the measurement chamber to the reference chamber. This was not accounted for in the volumes of the pressure chambers. In recent work we have instead changed the gas line configuration such that there is no longer a pressure difference between the sample and reference chambers.

*No changes have been made to the manuscript.*

**Figure 9: maybe color the points differently for the ramp up and down in NO2 concentration, so the outset is clear.**

This is a good suggestion to improve the clarity of Figure 9. The Figure will be updated in the manuscript to have different coloured points for the ramp up and down measurements.

*Figure 9 has been updated with square and round points for the ramp up and down, respectively. This allows the series to be differentiated without colour.*

**Line 439: The need to monitor the baseline drift negates the advantaged pointed out in the previous paragraph (Lines 416 -431).**

The authors respectfully disagree with this statement. There are two potential sources of the background change: gas absorption and photothermal effects in the optical elements in the interferometer.

Two-beam interferometers must in theory measure both changes or resort to another way of determining the concentration of gaseous species or employ e.g. a scrubber. We account for this with the reference chamber.

The absolute baseline must continuously be monitored in any PTI instrument as it arises primarily from light absorption by optical elements in the interferometer, which may change over time. This type of baseline change is different to that arising from the absorption due to gaseous species and changes thereof. Two beam configurations with glancing angle type configurations are rather sturdier in this respect. Our single beam configuration features a perfect overlap between the pump and the probe beams – they are the same laser beam, but this is also true for the effects in the optical elements. Hence the need to measure the background.

*No changes have been made to the manuscript.*

**Figure 11: Maybe this should be replaced with an Allan deviation plot which is appropriate to differentiate between short-term precession and long-term drift**

The authors feel that the standard deviation is more appropriate than the Allan deviation in expressing the uncertainty in PTI measurements. It is however acknowledged that presentation of the Allan deviation would better enable comparison to other measurement techniques. A plot of the Allan deviation using the same data as Figure 11 will be added to the supplementary information.

*A plot of the Allan deviation has been added to the supplementary information as Figure S8. A sentence has been added to the manuscript to direct the reader to Figure S8.*

**Line 492-495: Several photoacoustic absorption measurements use active charcoal scrubbers very effectively to remove gas-phase absorbers before measurement of the aerosol absorption.**

This is correct and a note to this effect will be added to the manuscript. It is however always advantageous to treat the aerosol as little as possible before measurement, in order to avoid changing any of its characteristics.

*The sentence in question was updated to: The MSPTI design also allows for the direct measurement of aerosol absorption in the presence of absorbing gases, which would normally require a complicated correction, a scrubber or secondary measurement of the gas absorption for other in-situ aerosol absorption measurements.*

**Reviewer 2**

**1. Introduction seems to be a little bit longer than what it ought to be. Is it important in this study to mention the vertical measurement of black carbon? If possible, please consider shrinking it into around 2 pages.**

The authors acknowledge that the introduction is a little lengthy. It will be reworked slightly for brevity.

*The introduction has been shortened by removing some non-critical sentences and rewording others for brevity.*

**2. As a matter of fact, one of the important things in the ambient measurement is the durability. Is the future experimental setup able to monitor the aerosol light absorption for longer than 24 hours?**

This is indeed a very important requirement for use of the instrument in ambient measurements. Recent tests have shown that the instrument can be operated for more than 24 hours without issue and the future instrument should be durable enough to measure for months on end to fulfil its function as an ambient monitoring instrument. A sentence will be added to the introduction of the manuscript to stress this point.

*The following sentences have been added to the Introduction of the manuscript: The durability and sensitivity of filter-based instruments have led to their employment in environmental monitoring stations.*

*And: Future improvement of the sensitivity and durability of the MSPTI is planned, enabling its use as a field monitoring instrument.*

**3. The reviewer is wondering if the authors checked the intensity of the laser power would be 50:50 after the beam-splitter. It would be great to leave a comment on the performance of the beam-splitter such as a plot of transmittance vs. wavelength. (Please see the graphs at https://www.thorlabs.com/newgrouppage9.cfm?objectgroup_id=914)**

Yes, the authors have investigated the splitting ratio of the beam splitter for various polarisations of the incoming laser beam and have seen a significant dependence of the splitting ratio on the polarisation. This was measured both using a power meter and by determining the contrast of the interferometer. We cannot comment on the wavelength dependence of the beam splitter as all of the measurements were performed at a single wavelength. Subsequent testing has shown that the splitting ratio and polarisation insensitivity is much improved for the 532 nm laser line beam splitter as compared with the broad-band version.

*No changes have been made to the manuscript.*

**4. Is it possible for the authors to say the power of laser in the current setup? The reviewer can only imagine it from the last paragraph of RESULT section (It must be less than 400 mW).**

The maximum power of the laser is 450 mW. The power employed in the study was 200 mW as at the time the cooling was insufficient to run the laser at higher powers for extended periods. The laser power employed in the study will be added to the experimental section of the manuscript.

*The following line of text was added to the Experimental section of the manuscript: The laser power was regulated at 200 mW in this study.*

**5. Line 63: One of the hallmarks of this study is to measure the optical property of light absorbing aerosols in an airborne state. In this sense, the reviewer recommends that the authors append a reference (Lee, 2019) to provide the drawback of any filter-based techniques.**

Thank you for the reference. It will be added to the introduction of the manuscript.

*The reference has been added to the specified line of the manuscript.*

**6. Line 238: Please make it sure whether the number of equation is correct or not. Equation 3? Or equation 5?**

 My apologies, the reviewer is correct, the sentence should indeed read Equation 5.

*This sentence has been updated in the manuscript.*

**7. Line 252: Please replace 'below a characteristic value' with 'shorter than a characteristic time'**

Thank you. The manuscript will be changed for this improved phrasing.

*The phrase 'below a characteristic value' has been replaced with the phrase 'shorter than a characteristic time' in the manuscript.*

**8. Line 286: Is the absolute filter HEPA-grade?**

Yes, the absolute filter is HEPA grade. The manuscript will be updated to reflect this.

*The phrase 'An absolute filter' has been replaced with the phrase 'A HEPA-grade absolute filter' in the manuscript.*

**9. Line 351-356: "PTI is an in situ light absorption ~ and the resultant PTI signal.". This is a general explanation about calibration, thus it is irrelevant to appear in Result section. The reviewer recommends that the sentences be moved to Experimental section, maybe at Line 349.**

The authors accept the recommendation and the lines in question will be moved to the end of the experimental section.

*The relevant sentences have been moved to the end of the Experimental section in the manuscript.*

**10. Line 376-380: This paragraph is unclear. Please reword the sentences so that potential readers may understand what it is.**

The authors intended to bring across the message that the heating curves were not observed to be linear, including for measurements made with heating periods in the so-called linear range seen in Figure 8. The shapes of the heating curves remained constantly non-linear in this range, however, thus leading to the linear relationship between the PTI signal and the heating time in this range. This section of text will be updated for clarity. Please see also replies to Anonymous Reviewer 1.

*The text:*

*It must be noted here that the existence of a linear relationship between the PTI signal and heating time does not imply that the heating curves themselves increase linearly with time. In fact, close examination of Figure 5 shows that this is very clearly not the case for the MSPTI. Instead the linear relationship between PTI signal and heating time implies that the shape of the heating curves remain similar for this range of heating times, the chosen evaluation of the PTI signal (e.g. linear fit or lock-in detection) is in good agreement with Eq. 6 and a calibration performed at one heating time can be directly transferred to measurements performed at a different heating time.*

*Has been replaced with the following text in the updated manuscript:*

*It must be noted that all of the heating curves recorded in this study were non-linear, even for heating times within the linear regime. The linear relationship between PTI signal and heating time for shorter heating times only implies that the shape of the heating curves remains constant for heating times within this range. Heating curves of a constant shape are evaluated consistently by the chosen linear-fit mechanism, thus allowing the transfer of calibration measurements from one heating time to another. Outside of the linear regime, calibration measurements cannot easily be transferred from one heating time to another, however measurements performed for an arbitrary heating time are still in good agreement with Eq. 6 as long as the shape of the heating curves remains constant (e.g. with concentration of the light absorbing species).*

**11. Line 456: The MAC and the filter multiple-scattering enhancement parameter were provided from the user's manual of AE33. Is it possible for the authors to comment on how 13.14 m$^2$/g and 1.57 were derived for the MAC and the multiple scattering parameter, respectively?**

Filter photometers are calibrated using the mass attenuation cross-section (Gundel et al., 1984). The mass attenuation cross-section is a product of the mass absorption cross-section and the filter multiple-scattering parameter, using the parameterization of Weingartner et al. (2003). Drinovec et al. (2015) have determined the C for the AE33 filter (at that time) relative to the value from Weingartner et al. (2003). Filter photometer response to a complex sample with a high SSA is more complicated (Lee, 2019) – the scattering of the sample affects the measurements (Weingartner et

al., 2003; Arnott et al., 2005) and this cross-sensitivity to scattering affects the measurement. This is often measured as a change in the effective (apparent) multiple-scattering parameter, that is the slope between the reference absorption measurement and the filter photometer. We do not observe this effect as the SSA of our aerosol samples is very low.

*No changes have been made to the manuscript.*

**References Lee J. Performance Test of MicroAeth® AE51 at Concentrations Lower than 2 µg/m3 in Indoor Laboratory. Applied Sciences. 2019, 9(13), 2766.**

**Reviewer 3**

**Page 6, line 11. What was the criteria for selecting I_low and I_high? All that is stated is that the laser "is modulated between two sufficiently different intensity levels." Certainly there is a lower limit below which the ability to lock on to quadrature would be compromised. On the other hand, the larger the difference, would favor signal detection.**

Yes, the reviewer is completely correct and a larger difference between I_low and I_high leads to an increase in signal. The initial selection of I_low was determined by the accuracy at which the signal during the cooling (or low) phase could be determined. As this data was used to lock quadrature and provide some qualitative indication of the signal during the cooling phase, a minimum laser power of 20 mW was required to provide acceptable signal to noise. It should however be noted, that it is possible to lock quadrature with only the heating phase (or high) signals, thus making it possible for I_low to be zero, if no information about the cooling phase is required. However, with the AOM we have so far only been able to achieve a 20x reduction in the laser intensity in the main beam and therefore we have not been able to test this without reconfiguring the outputs of the AOM.

Additional information will be added to the manuscript to explain this.

*The following sentences were added to the manuscript to better explain the choice of I_low (I_high is fixed by the maximum laser power available): A larger intensity difference between the levels leads to an increased PTI signal, however signal to noise limitations restrict the choice of $I_{low}$ in the case*

*that measurements are made during the laser low phase. In this study $I_{low}$ was set to $\frac{1}{10}I_{high}$, which allowed a qualitative indication of the signal response during the cooling phase.*

**Page 8, lines 8 and 9: the use of "solid" and "external" noise sources is not very descriptive. Why not call this noise sources what they actually are: mechanical (vibrational) noise and acoustic noise. For those unfamiliar with PTI or, more generally interferometry, referring to a noise source as "solid" or "external" is a bit nebulous.**

The authors agree on this point and the manuscript will be updated to make it clearer in this respect.

*The sentences in question have been updated in the manuscript to: As both the reference and measurement beams are incident on the beam splitter and mirror, the effects of mechanical (vibrational) noise are reduced when compared to standard Michelson or Mach-Zehnder designs. The insensitivity to mechanical noise is not as complete as for the Jamin design, as the two optical elements are able to move with respect to each other, but the design does allow for flexibility in the design of the aerosol chamber.*

*Furthermore, the wording changes have been applied throughout the manuscript.*

**Page 8, Lines 15-17. The MSPTI utilizes a reference channel that samples filtered air - a necessary condition for the single-beam configuration to work. What is the impact of a sample containing a mixture of light absorbing and non-light absorbing particles at high concentrations, as might be encountered in a biomass burning event, where the refractive index (RI) of the particles could contribute to the sample ensemble RI but whose contributions would not be present in the particle filtered sample? What are thermal lensing implications under these conditions? In a two-beam PTI, the sample and reference arms probe the same particle-laden air simultaneously thereby enabling common mode rejection for such conditions.**

This is a very interesting question and one that will require some further study to experimentally validate. In theory, PTI measurements are only sensitive to changes in the measured refractive index at the modulation frequency. This means that static differences in refractive index, as well as slow changes in the refractive index (even relative changes) should not affect the measured signal, regardless of the source. Even at very high concentrations, where the RIs of the various particles contribute in a meaningful way to the ensemble, the authors do not believe that any significant artefacts due to a static difference in RI will be present. At such high concentrations however, light attenuation by the sample will be large and therefore the interferometric contrast will suffer, reducing the accuracy of the measurements. We have not tested the MSPTI at sufficiently high concentrations to investigate this, but there is almost certainly an upper concentration limit beyond which the measurements are no longer valid.

The implications for thermal lensing are the same as for the two-beam PTI configurations. A thermal lens will be formed in the sample beam, which is the equivalent of the sample beam / pump beam thermal lens in the two-beam PTI configuration (with some differences due to the beam geometries). In the absence of light absorbing gases, no thermal lens will be formed in the reference beam, just as for a two-beam configuration. It is however slightly different in the case where an absorbing gas is present in the reference chamber. Under these conditions a thermal lens will be formed in the reference beam of the MSPTI, whereas none would be formed in the two-beam PTI. However, since the absorption of the gas is the same in both sample and reference chambers, the contribution to the thermal lens from the gas absorption in both chambers should also be equal.

*No changes have been made to the manuscript.*

**Page 10, line 4. The authors are encouraged to cite Lack et al. (2006) here. This paper is already listed in their citations.**

The citation will be added to this line of the manuscript.

*The citation has been added to the manuscript at the specified line.*

**Page 11, line 16. The authors are encouraged to merge Figures 5 and 8. In a lot of ways, Figure 8 is far more informative as it beautifully captures how decreasing the modulation frequency - increasing the heating period - brings about significant departure from linearity due to energy diffusion outside the probe region.**

This is unfortunately confusing, however Figures 5 and 8 don't show the same data or effect. Even the heating curves for the data points in the linear dependence region of Figure 8 are not themselves linear. The linear dependence of the PTI signal on the heating period in Figure 8 only shows that the shape of the heating curves has remained consistent within this range. The discrepancy between the heating curve and the linear fit outside of this range increases with increasing heating period, causing the nonlinearity of the PTI signal as a function of the heating period (Figure 8).

*No changes have been made to the manuscript.*

**Page 11, Line 22. The sentence "If the deviation from linearity of the PTI signal with heating time due to heat loss out of the measurement volume could be excluded from the measurement, then both the measured signal and, by extension, the sensitivity determined from calibration measurements would be considerably higher." This is a very awkwardly worded sentence. I believe that the authors simply trying to say that the non-linear signal due to diffusional loss of heat outside the probe volume suppresses measurement of the total amount of energy deposited into the system. If so, please clarify. [As an aside from purely physics interest, this raises an interesting question with respect to a two-beam PTI: if the probe volume was configured to be slightly larger than the pump volume would this enable the 2-beam configuration to "delay" the onset of the departure from linearly and, in so doing, improve performance at lower modulation frequencies? ]**

Yes, thank you for helping to clarify this section of text. The manuscript will be updated to clarify this point.

The saturation should occur at longer times by increasing the diameter of the probe beam, assuming no other cause of non-linearity becomes dominant. However, the signal strength is the intensity weighted average of the phase shift across the beam cross-section. The signal at shorter heating times will be reduced due to the time it takes for the heat from the absorption process to be conducted into the entire probe beam volume.

*The sentence highlighted by the reviewer has been replaced with the one suggested.*

**Page 11, line 36. It seems to this reviewer that the two time series traces should be switched. Ideally, the authors should first show that their system can indeed detect NO2 (currently the right most trace) and THEN show how well their system does at removing the NO2 signal (currently the left most trace). The actual time stamps is immaterial here. This is a stylistic comment.**

Yes, the authors concur that this would be preferential, both in a scientific as well as aesthetic manner. The figure will be updated in the manuscript.

*Figure 10 has been updated to present the two time traces in the opposite order. Now the trace using the calibration flow configuration is presented first and then the trace using the measurement flow configuration is presented.*

**Page 12, line 2: The authors are reminded that there are chemical "denuders" for removing molecular species such as NO2 via MnO2.**

This is correct. The authors were trying to make the point that it is possible for the MSPTI to measure aerosol particle absorption in the presence of absorbing gases without the need to modify the sample. The manuscript will be updated to clarify this point and the availability of denuders.

*The sentence was updated to read: This is a significant advantage over previous PTI designs, which rely on either periodic measurements of the background gas absorption, $NO_2$ denuders or the measurements of other sensors to determine the aerosol absorption from the total absorption.*

**Page 12, Comparison with Aethalometer. Philosophically, this reviewer has major concerns about the underlying assumption of a constant mass absorption cross-section (MAC) for black carbon (BC) in order to report an equivalent black carbon (eBC) concentration. There are a plethora of studies showing that the BC MAC (at 550 nm, for example) can vary from ~7.5 m2/g for uncoated BC particles to 13-15 m2/g for coated particles. This reviewer understands that the Aethalometer reports a eBC value and that the authors are comparing their instrument to the Aethalometer. While this comment is well-beyond the scope of this present paper, one potential (and easy) solution that the authors might consider, is to compare absorption coefficients instead of mass concentrations - after all, this is what both instruments fundamentally measure.**

This is correct. The mass concentrations were only intended for comparison with other measurements and to give aerosol scientists a metric to better understand the measured absorption (i.e. mass concentrations). The manuscript will be updated to focus more on absorption and use eBC as a secondary metric to clarify this point.

*The text has been updated to focus on the measurement of absorption, with equivalent mass concentrations of black carbon (eBC) provided as a secondary metric for comparison and understanding purposes. The figures have also been updated to present the data in terms of absorption units ($Mm^{-1}$).*

**Page 23, Figure 11. Is the departure observed in the variation from tˆ-1/2 due to the active quadrature lock feedback circuit? Also a more meaningful metric to the aerosol community would be an Allan variance plot of the absorption coefficient.**

As the quadrature lock circuit was operating at a frequency of 1 Hz or below, one source of the deviation from the inverse square ideal line could be partially due to the constant adjustments from this circuit. The main contribution however seems to be the low frequency drifts in the baseline.

The authors feel that the standard deviation is more appropriate than the Allan deviation in expressing the uncertainty in PTI measurements. It is however acknowledged that presentation of the Allan deviation would better enable comparison to other measurement techniques. A plot of the Allan deviation using the same data as Figure 11 will be added to the supplementary information.

*A plot of the Allan deviation has been added to the supplementary information as Figure S8.*

Other changes have been made to the manuscript to improve clarity (see the attached marked up document for details

[revised manuscript text omitted]